# RKHS-SHAP: Shapley Values for Kernel Methods

**Siu Lun Chau**
Department of Statistics
University of Oxford

**Robert Hu**
Amazon[†]
London

**Javier Gonzalez**
Microsoft Research
Cambridge

**Dino Sejdinovic**
School of Computer and Mathematical Sciences[†]
University of Adelaide

## Abstract

Feature attribution for kernel methods is often heuristic and not individualised for each prediction. To address this, we turn to the concept of Shapley values (SV), a coalition game theoretical framework that has previously been applied to different machine learning model interpretation tasks, such as linear models, tree ensembles and deep networks. By analysing SVs from a functional perspective, we propose RKHS-SHAP, an attribution method for kernel machines that can efficiently compute both *Interventional* and *Observational Shapley values* using kernel mean embeddings of distributions. We show theoretically that our method is robust with respect to local perturbations - a key yet often overlooked desideratum for consistent model interpretation. Further, we propose *Shapley regulariser*, applicable to a general empirical risk minimisation framework, allowing learning while controlling the level of specific feature's contributions to the model. We demonstrate that the Shapley regulariser enables learning which is robust to covariate shift of a given feature and fair learning which controls the SVs of sensitive features.

## 1 Introduction

Machine learning model interpretability is critical for researchers, data scientists, and developers to explain, debug and trust their models and understand the value of their findings.

A typical way to understand model performance is to attribute importance scores to each input feature [5]. These scores can be computed either for an entire dataset to explain the model's overall behaviour (global) or compute individually for each single prediction (local).

Understanding feature importances in reproducing kernel Hilbert space (RKHS) methods such as kernel ridge regression and support vector machines often require the study of kernel lengthscales across dimensions [44, Chapter 5]. The larger the value, the less relevant the feature is to the model. Albeit straightforward, this approach comes with three shortcomings: (1) It only provides global feature importances and cannot be individualised to each single prediction.

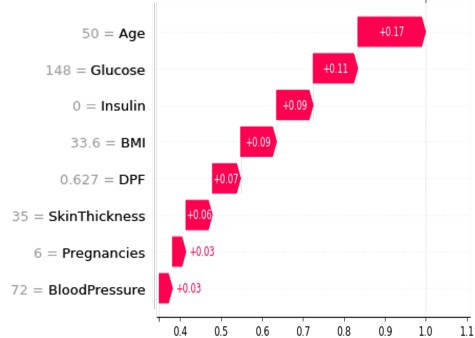

Figure 1: An example of RKHS-SHAP providing local explanations to why a kernel logistic model predicts this patient to be diabetic. [11]. RKHS-SHAP provides a more granular level of explanation than studying lengthscales across dimensions.

---

[†] Work mainly done while the authors were with the Department of Statistics, University of Oxford

36th Conference on Neural Information Processing Systems (NeurIPS 2022).

This explanation is limited as global importance does not necessarily imply local importance [33]). In safety critical domain such as medicine, understanding individual prediction is arguably more important than capturing the general model performance. See Fig 1 for an example of local explanation. (2) The tuning of lengthscales often requires a user-specified grid of possible configurations and is selected using cross-validations. This pre-specification thus injects substantial amount of human bias to the explanation task. (3) Lengthscales across kernels acting on different data types, such as binary and continuous variables, are difficult to compare and interpret.

To address this problem we turn to the Shapley value (SV) [35] literature, which has become central to many model explanation methods in recent years. The Shapley value was originally a concept used in game theory that involves fairly distributing credits to players working in coalition. Štrumbelj and Kononenko [40] were one of the first to connect SV with machine learning explanations by casting predictions as coalition games, and features as players. Since then, a variety of SV based explanation models were proposed. For example, LINEARSHAP [40] for linear models, TREESHAP [24] for tree ensembles and DEEPSHAP [23] for deep networks. Model agnostic methods such as DATA-SHAPLEY [15], SAGE [9] and KERNELSHAP [2] [23] were also proposed. However, to the best of our knowledge, an SV-based local feature attribution framework suited for kernel methods has not been proposed.

While one could still apply model-agnostic KERNELSHAP on kernel machines, we show that by representing distributions as elements in the RKHS through kernel mean embeddings [38, 27], we can compute Shapley values more efficiently by circumventing the need to sample and estimate an exponential amount of densities required to compute the value functions, an essential component for Shapley value computation. We call this approach RKHS-SHAP to distinguish it from KERNELSHAP. Through the lens of RKHS, we study Shapley values from a functional perspective and prove that our method is robust with respect to local perturbations under mild assumptions, which is an important yet often neglected criteria for explanation models as discussed in Hancox-Li [17]. In addition, a *Shapley regulariser* based on RKHS-SHAP is proposed for the empirical risk minimisation framework, allowing the modeller to control the degree of feature contribution during the learning. We also discuss its application to robust learning to covariate shift of a given feature and fair learning while controlling contributions from sensitive features. We summarise our contributions below:

**1.** We propose RKHS-SHAP, a model specific algorithm to compute Shapley values efficiently for kernel methods by circumventing the need to sample and fit from an exponential number of densities.

**2.** We prove that the corresponding Shapley values are robust to local perturbations under mild assumptions, thus providing consistent explanations for the kernel model.

**3.** We propose a *Shapley regulariser* for the empirical risk minimisation framework, allowing the modeller to control the degree of feature contribution during the learning.

The paper is outlined as follows: In section 2 we provide an overview of Shapley values and kernel methods. In section 3 we introduce RKHS-SHAP and show robustness of the algorithm. *Shapley regulariser* is introduced in section 4. Section 5 provides extensive experiments. We conclude our work in section 6.

## 2 Background Materials

**Notation.** We denote $X, Y$ as random variables (rv) with distribution $p(X, Y)$ taking values in the $d$-dimensional instance space $\mathcal{X} \subseteq \mathbb{R}^d$ and the label space $\mathcal{Y}$ (could be in $\mathbb{R}$ or discrete) respectively. We use $D = \{1, ..., d\}$ to denote the feature index set of $X$ and $S \subseteq D$ to denote the subset of features of interests. Lower case letters are used to denote observations from corresponding rvs.

### 2.1 The Shapley Value

The Shapley value was first proposed by Shapley [35] to allocate performance credit across coalition game players in the following sense: Let $\nu : \{0, 1\}^d \to \mathbb{R}$ be a *coalition game* that returns a score for each coalition $S \subseteq D_g$, where $D_g = \{1, ..., d\}$ represents a set of players. Assuming the grand

---

[2]The kernel in KERNELSHAP refers to the estimation procedure is not related to RKHS kernel methods.

coalition $D_g$ is participating and one wished to provide the $i^{th}$ player with a fair allocation of the total profit $\nu(D_g)$, how should one do it? Surely this is related to each player's *marginal contribution* to the profit with respect to a coalition $S$, i.e. $\nu(S \cup i) - \nu(S)$. Shapley [35] proved that there exists a *unique* combination of marginal contributions that satisfies a set of favourable and fair game theoretical axioms, commonly known as *efficiency, null player, symmetry* and *additivity*. This unique combination of contributions is later denoted as the *Shapley value*. Formally, given a coalition game $\nu$, the Shapley value for player $i$ is computed as the following,

$$\phi_i(\nu) = \frac{1}{d} \sum_{S \subseteq D_g \backslash \{i\}} \binom{d-1}{|S|}^{-1} \Big( \nu(S \cup i) - \nu(S) \Big). \tag{1}$$

**Choosing $\nu$ for ML explanation**   In recent years, the Shapley value concept has become popular for feature attribution in machine learning. SHAP [23], SHAPLEY EFFECT [37], DATA-SHAPLEY [15] and SAGE [9] are all examples that cast model explanations as coalition games by choosing problem-specific value functions $\nu$. Denote $f : \mathcal{X} \to \mathcal{Y}$ as the machine learning model of interest. Value functions for local attribution on observation $x$ often take the form of the expectation of $f$ with respect to some reference distribution $r(X_{S^c} \mid X_S = x_S)$, where $S \subseteq D$ is some coalition of features in analogous to the game theory setting, such that:

$$\nu_{x,S}(f) = \mathbb{E}_{r(X_{S^c}|X_S=x_S)}[f(\{x_S, X_{S^c}\})], \tag{2}$$

where $\{x_S, X_{S^c}\}$ denotes the concatenation of the arguments. We wrote $f$ as the main argument of $\nu$ to highlight its interpretation as a functional indexed by local observation $x$ and coalition $S$. When $r$ is set to be marginal distribution, i.e $r(X_{S^c} \mid X_S = x_S) = p(X_{S^c})$, the value function is denoted as the *Interventional value function* by Janzing et al. [20]. *Observational value function* [13], on the other hand, set the reference distribution to be a conditional distribution $p(X_{S^c} \mid X_S = x_S)$. Other choices of reference distributions will lead to Shapley values with specific properties, e.g., better locality of explanations [14] or incorporating causal knowledge [18]. In this work we shall restrict our attention to marginal and conditional cases as they are the two most commonly adopted choices in the literature.

**Definition 1.** *Given model $f$, local observation $x$ and a coalition set $S \subseteq D$, the Interventional and Observational value functions are denoted by $\nu_{x,S}^{(I)}(f) := \mathbb{E}_{X_{S^c}}[(f(\{x_S, X_{S^c}\})]$ and $\nu_{x,S}^{(O)}(f) := \mathbb{E}_{X_{S^c}}[f(\{x_S, X_{S^c}\}) \mid X_S = x_S]$.*

The right choice of $\nu$ has been a long-standing debate in the community. While Janzing et al. [20] argued from a causal perspective that $\nu_{x,f}^{(I)}$ is the correct notion to represent missingness of features in an explanation task, Frye et al. [13] argued that computing marginal expectation ignores feature correlation and leads to unrealistic results since one would be evaluating the value function outside the data-manifold. This controversy was further investigated by Chen et al. [6], where they argued that the choice of $\nu$ is *application dependent* and the two approaches each lead to an explanation that is either *true to the model* (marginal expectation) or *true to the data* (conditional expectation). When the context is clear, we denote the Shapley value of the $i^{th}$ feature of observation $x$ at $f$ as $\phi_{x,i}(f)$ and use a superscript to indicate whether it is *Interventional* $\phi_{x,i}^{(I)}(f)$ or *Observational* $\phi_{x,i}^{(O)}(f)$.

**Computing Shapley values.**   While Shapley values can be estimated directly from Eq. (1) using a sampling approach [40], Lundberg and Lee [23] proposed KERNELSHAP, a more efficient algorithm for estimating Shapley values in high dimensional feature spaces by casting Eq. (1) as a weighted least square problem. Similar to LIME [33], for each data $x$, model $f$, and feature coalition $S$, KERNELSHAP places a linear model $u_x(S) = \beta_{x,0} + \sum_{i \in S} \beta_{x,i}$ to explain the value function $\nu_{x,S}(f)$, which corresponds to solving the following regression problem: $\min_{\beta_{x,0},...,\beta_{x,d}} \sum_{S \subseteq D} w(S)(u_x(S) - \nu_{x,S}(f))^2$, where $w(S) = \frac{d-1}{\binom{d}{|S|}|S|(d-|S|)}$ is a carefully chosen weighting such that the regression coefficients recover Shapley values. In particular, one set $w(\varnothing) = w(D) = \infty$ to effectively enforce constraints $\beta_{x,0} = \nu_{x,\varnothing}(f)$ and $\sum_{i \in D} \beta_{x,i} = \nu_{x,D}(f) - \nu_{x,\varnothing}(f)$. Denoting each subset $S \subseteq D$ using the corresponding binary vector $\mathbf{z} \in \{0,1\}^d$, and with an abuse of notation by setting $\nu_{\cdot,\mathbf{z}} := \nu_{\cdot,S}$ and $w(\mathbf{z}) := w(S)$ for $S = \{j : \mathbf{z}[j] = 1\}$, we can express the Shapley values $\boldsymbol{\beta}_x := [\beta_{x,0},...,\beta_{x,d}]$ as $\boldsymbol{\beta}_x = (Z^\top W Z)^{-1} Z^\top W \mathbf{v}_x$ where $Z \in \mathbb{R}^{2^d \times d}$ is the binary matrices with columns $\{\mathbf{z}_i\}_{i=1}^{2^d}$, $W$ is the diagonal matrix with entries $w_{ii} = w(\mathbf{z}_i)$ and

$\mathbf{v}_x := \{\nu_{x,\mathbf{z}_i}(f)\}_{i=1}^{2^d} \in \mathbb{R}^{2^d \times 1}$ the vector of evaluated value functions, which is often estimated using sampling and data imputations. We shall explain the pathology of this approach in detail later in Section 3. In practice, instead of evaluating at all $2^d$ combinations, one would subsample the coalitions $z \sim w(z)$ for computational efficiency [8].

**Model specific Shapley methods.** KERNELSHAP provides efficient model-agnostic estimations of Shapley values. However, by leveraging additional structural knowledge about specific models, one could further improve computational performance. This leads to a variety of model-specific approximations, most of which relies on utilising their specific structure to speed up computation of value functions. For example, LINEARSHAP [40] explain linear models using model coefficients directly. TREESHAP [24] provides an exponential reduction in complexity compared to KERNELSHAP by exploiting the tree structure. DEEPSHAP [23], on the other hand, combines DEEPLIFT [36] with Shapley values and uses the compositional nature of deep networks to improve efficiencies. However, to the best of our knowledge, a kernel method specific Shapley value approximation has not been studied. Later in Section 3, we will show that, under a mild structural assumption on the RKHS, kernel methods can be used to speed up the computation in KERNELSHAP by estimating value functions analytically, thus circumventing the need for estimating and sampling from an exponential number of densities.

**Related work on kernel-based Shapley methods.** Da Veiga [10]'s work on tackling global sensitive analysis by proposing the kernel-based maximum mean discrepancy as value function, is conceptually most similar to ours. However, there are multiple key differences in our contributions. Firstly, their method is designed for global explanation, while ours is for local. Secondly, similar to interventional SV, they do not consider any conditional distributions, thus leading to completely different estimation procedures and thus novelty. Lastly, their method is on understanding the input/outputs relationship of a numerical simulation model, while ours focuses on understanding specific RKHS models learnt from a machine learning task, e.g. kernel ridge regression and kernel logistic regression.

## 2.2 Kernel Methods

Kernel methods are one of the pillars of machine learning, as they provide flexible yet principled ways to model complex functional relationships and come with well-established statistical properties and theoretical guarantees.

**Empirical Risk Minimisation.** Recall in the supervised learning framework, we are learning a function $f : \mathcal{X} \to \mathcal{Y}$ from a hypothesis space $\mathcal{H}$, such that given a training set $(\mathbf{x}, \mathbf{y}) = \{(x_i, y_i)\}_{i=1}^n$ sampled identically and independently from $p$, the following empirical risk is minimised: $f^* = \arg\min_{f \in \mathcal{H}} \frac{1}{n} \sum_{i=1}^n \ell(y_i, f(x_i)) + \lambda_f \Omega(f)$, where $\ell : \mathcal{Y} \times \mathcal{Y} \to \mathbb{R}$ is the loss function, $\Omega : \mathcal{H} \to \mathbb{R}$ a regularisation function and $\lambda_f$ a scalar controlling the level of regularisation. Denote $k : \mathcal{X} \times \mathcal{X} \to \mathbb{R}$ a positive definite kernel with feature map $\psi_x$ for input $x \in \mathcal{X}$ and $\mathcal{H}_k$ the corresponding RKHS. If we pick $\mathcal{H}_k$ as our hypothesis space, then the *Representer theorem* [39] tells us that the optimal solution takes the form of $f^* = \sum_{i=1}^n \alpha_i k(\cdot, x_i) = \Psi_\mathbf{x} \boldsymbol{\alpha}$, where $\Psi_\mathbf{x} = [\psi_{x_1} \dots \psi_{x_n}]$ is the feature matrix defined by stacking feature maps along columns. If $\ell$ is the squared loss then the above optimisation is known as kernel ridge regression and $\boldsymbol{\alpha}$ can be recovered in closed form $\boldsymbol{\alpha} = (\mathbf{K}_{\mathbf{xx}} + \lambda_f I)^{-1} \mathbf{y}$, where $\mathbf{K}_{\mathbf{xx}} = \Psi_\mathbf{x}^\top \Psi_\mathbf{x}$ is the kernel matrix. If $\ell$ is the logistic loss, then the problem is known as kernel logistic regression, and $\boldsymbol{\alpha}$ can be obtained using gradient descent.

**Kernel embedding of distributions.** An essential component for RKHS-SHAP is the embedding of both marginal and conditional distribution of features into the RKHS [38, 27], thus allowing one to estimate the value function analytically. Formally, the kernel mean embedding (KME) of a marginal distribution $P_X$ is defined as $\mu_X := \mathbb{E}_X[\psi_X] = \int_{\mathcal{X}} \psi_x dP_X(x)$ and the empirical estimate can be obtained as $\hat{\mu} := \frac{1}{n} \sum_{i=1}^n \psi_{x_i}$. Furthermore, given another kernel $g : \mathcal{Y} \times \mathcal{Y} \to \mathbb{R}$ with feature map $\psi_Y$ of RKHS $\mathcal{H}_g$, the conditional mean embedding (CME) of the conditional distribution $P_{Y|X=x}$ is defined as $\mu_{Y|X=x} := \mathbb{E}[\psi_Y | X = x] = \int_{\mathcal{Y}} \psi_y dP_{Y|X=x}(y)$.

One way to understand CME is to view it as an evaluation of a vector-valued(VV) function $\mu_{Y|X} : \mathcal{X} \to \mathcal{H}_g$ such that $\mu_{Y|X}(x) = \mu_{Y|X=x}$, which minimises the following risk function $\mathbb{E}_{p(X,Y)}[||\psi_Y - \mu_{Y|X}(X)||_{\mathcal{H}_g}^2]$ [16]. Let $\mathcal{L}(\mathcal{H}_g)$ be the space of bounded linear operators from $\mathcal{H}_g$ to

itself. Denote $\Gamma_x : \mathcal{X} \times \mathcal{X} \to \mathcal{L}(\mathcal{H}_g)$ as the operator-valued kernel such that $\Gamma_x(x, x') = k(x, x')\mathbf{1}$ with $\mathbf{1}$ the identity operator on $\mathcal{H}_g$. We denote $\mathcal{H}_{\Gamma_x}$ as the corresponding vector-valued RKHS. By utilising the VV-Representer theorem [26], we could minimises the following empirical risk:

$$\hat{\mu}_{Y|X} = \underset{\mu_{Y|X} \in \mathcal{H}_{\Gamma_x}}{\arg\min} \sum_{i=1}^{n} ||\psi_{y_i} - \mu_{Y|X}(x_i)||_{\mathcal{H}_g}^2 + n\eta||\mu_{Y|X}||_{\Gamma_x}^2$$

where $\eta > 0$ is a regularisation parameter. This leads to the following empirical estimate of the CME, i.e., $\hat{\mu}_{Y|X} = \Psi_{\mathbf{y}}(\mathbf{K_{xx}} + n\eta I)^{-1}\Psi_{\mathbf{x}}^\top$, where $\Psi_{\mathbf{y}} := [\psi_{y_1}...\psi_{y_n}]$ and $\Psi_{\mathbf{x}} := [\psi_{x_1}...\psi_{x_n}]$ are feature matrices. Intuitively, this essential turns CME estimation to a regression problem from $\mathcal{X}$ to the vector-valued labels $\psi_Y$. Please see Micchelli and Pontil [26] and Grünewälder et al. [16] for further discussions on vector-valued RKHSs and CMEs. In fact, when using finite-dimensional feature maps, such as in the case with running Random Fourier Features [32] and Nyström methods [45] for scalability, one could reduce the computational complexity of evaluating empirical CME from $\mathcal{O}(n^3)$ to $\mathcal{O}(b^3) + \mathcal{O}(b^2 n)$ [27] where $b$ is the dimension of the feature map and often can be chosen much smaller than $n$ [21].

# 3 RKHS-SHAP

While KERNELSHAP is model agnostic, by restricting our attention to the class of kernel methods, faster Shapley value estimation can be derived. We assume our RKHS takes a tensor product structure, i.e, $\mathcal{H}_k = \bigotimes_{i=1}^{d} \mathcal{H}_{k^{(i)}}$, where $k^{(i)}$ is the kernel for each dimension $i \in D$. This structural assumption allows us to decompose the value functionals into tensor products of embeddings and feature maps, thus we can estimate them analytically, as later shown in Prop. 2. Tensor product RKHSs are commonly used in practice, as they preserve universalities of kernels from individual dimension [42], thus providing a rich function space. Note that this assumption is not essential within our framework. Namely, for a non-product kernel, one can still evaluate the value functions using tools from conditional mean embeddings and utilise our interpretability pipeline without conditional density estimation. We show this in Appendix C. In the following, we will lay out the disadvantage of existing sampling and data imputation approach and show that by estimating the value functionals as elements in the RKHS, we can circumvent the need for learning and sampling from an exponential number of conditionals densities – thus improving the computational efficiency in the estimation.

**Estimating value functions by sampling.** Estimating the Observational value function $\nu_{x,S}^{(O)}(f)$ is typically much harder than the Interventional value function $\nu_{x,S}^{(I)}(f)$ as it requires integration with respect to the unknown conditional density $p(X_{S^c} \mid X_S)$. Therefore, estimating OSVs often boils down to a two-stage approach: (1) Conditional density estimation and (2) Monte Carlo averaging over imputed data, as shown in Aas et al. [1], where they considered using multivariate Gaussian and Gaussian Copula for density estimation. Recently, an alternate way to estimate observational value functions is proposed by Frye et al. [13], where they formulate the estimation as a regression problem and compute the value function using a masked neural network directly without making any distributional assumption. This method shares conceptual similarities to ours but uses very different tools for the estimation. We highlight such differences in the appendix B.

Once the conditional density function $p(X_{S^c} \mid X_S)$ for each $S \subseteq D$ is estimated, the observational value function at the $i^{th}$ observation $x_i$ can then be computed by taking averages of $m$ Monte Carlo samples from the estimated conditional density, i.e. $\frac{1}{m} \sum_{j=1}^{m} f(\{x_{iS}, x_{jS^c}\})$ where $\{x_{iS}, x_{jS^c}\}$ is the concatenation of $x_{iS}$ with the $j^{th}$ sample $x_{jS^c}$ from $p(X_{S^c} | X_S = x_{iS})$. Note further that the Monte Carlo samples cannot be reused for another observation $x_k$ as their conditional densities are different. In other words, $n \times m$ Monte Carlo samples are required for each coalition $S$ if one wishes to compute Shapley values for all $n$ observations. This is clearly not desirable. In the spirit of Vapnik's principle[3], as our goal is to estimate conditional expectations that lead to Shapley values, we are not going to solve a harder and more general problem of conditional density estimation as an intermediate step, but instead utilise the arsenal of kernel methods to estimate the conditional expectations directly. Further discussion on comparing complexity of RKHS-SHAP with density estimation methods can be found in Appendix A.

---

[3]*When solving a problem, try to avoid solving a more general one as an intermediate step.* [43, Section 1.9]

**Estimating value functions using mean embeddings.** If our model $f$ lives in $\mathcal{H}_k$, both the marginal and conditional expectation can be estimated analytically without any sampling or density estimation. We first show that the Riesz representations [30] of both *Interventional* and *Observational value functionals* exist and are well-defined in $\mathcal{H}_k$. In the following, for simplicity, we will denote the functional and its corresponding Riesz representer using the same notation. For example, we will write $\nu_{x,S}(f) = \langle f, \nu_{x,S} \rangle_{\mathcal{H}_k}$ when the context is clear. Given a vector of $n$ instances $\mathbf{x}$, we denote the corresponding vector of value functions as $\nu_{\mathbf{x},S}(f) = \{\nu_{x_i,S}(f)\}_{i=1}^n$. All proofs of this paper can be found in the Appendix D.

**Proposition 2** (Riesz representations of value functionals). *Denote $k$ as the product kernel of $d$ bounded kernels $k^{(i)} : \mathcal{X}^{(i)} \times \mathcal{X}^{(i)} \to \mathbb{R}$, where $\mathcal{X}^{(i)}$ is the domain of the $i^{th}$ feature for $i \in D$. Riesz representations of the Interventional and Observational value functionals then exist and can be written as $\nu_{x,S}^{(I)} = \psi_{x_S} \otimes \mu_{X_{S^c}}$ and $\nu_{x,S}^{(O)} = \psi_{x_S} \otimes \mu_{X_{S^c}|X_S=x_S}$, where $\psi_{x_S} := \bigotimes_{i \in S} \psi_{x^{(i)}}$, $\mu_{X_{S^c}} := \mathbb{E}[\bigotimes_{i \in S^c} \psi_{x^{(i)}}]$ and $\mu_{X_{S^c}|X_S=x_S} := \mathbb{E}[\bigotimes_{i \in S^c} \psi_{x^{(i)}} | X_S = x_S]$.*

The corresponding finite sample estimators $\hat{\nu}_{x,S}^{(I)}$ and $\hat{\nu}_{x,S}^{(O)}$ are then obtained by replacing the corresponding KME and CME components with their empirical estimators. As a result, given $f^* = \Psi_{\mathbf{x}}\boldsymbol{\alpha}$ trained on dataset $(\mathbf{x}, \mathbf{y})$, Prop. 2 allows us to estimate the value functionals analytically since $\hat{\nu}_{x,S}^{(I)}(f^*) = \langle f^*, \psi_{x_S} \otimes \hat{\mu}_{X_{S^c}} \rangle$ and $\hat{\nu}_{x,S}^{(O)}(f^*) = \langle f^*, \psi_{x_S} \otimes \hat{\mu}_{X^{S^c}|X_S=x_S} \rangle$. This corresponds to the direct non-parametric estimators of value functions given in the following proposition, which circumvent the need for sampling or density estimation.

**Proposition 3.** *Given $\mathbf{x}' \in \mathbb{R}^{n'}$ a vector of instances and $f = \Psi_{\mathbf{x}}\boldsymbol{\alpha}$, the empirical estimates of the functionals can be computed as, $\hat{\nu}_{\mathbf{x}',S}^{(I)}(f) = \boldsymbol{\alpha}^\top \mathcal{K}_{\mathbf{x}',S}^{(I)}, \hat{\nu}_{\mathbf{x}',S}^{(O)}(f) = \boldsymbol{\alpha}^\top \mathcal{K}_{\mathbf{x}',S}^{(O)}$, respectively, where $\mathcal{K}_{\mathbf{x}',S}^{(I)} = \mathbf{K}_{\mathbf{x}_S \mathbf{x}'_S} \odot \frac{1}{n} \text{diag}(\mathbf{K}_{\mathbf{x}_{S^c}\mathbf{x}_{S^c}}^\top \mathbf{1_n})\mathbf{1_n}\mathbf{1_{n'}}^\top$ and $\mathcal{K}_{\mathbf{x}',S}^{(O)} = \mathbf{K}_{\mathbf{x}_S \mathbf{x}'_S} \odot \Xi_S \mathbf{K}_{\mathbf{x}_S \mathbf{x}'_S}$, $\mathbf{1_n}$ is the all-one vector with length $n$, $\odot$ the Hadamard product and $\Xi_S = \mathbf{K}_{\mathbf{x}_{S^c}\mathbf{x}_{S^c}}(\mathbf{K}_{\mathbf{x}_S \mathbf{x}_S} + n\eta I)^{-1}$.*

Finally, to obtain the Shapley values with these value functions, we deploy the same least square approach as KERNELSHAP.

**Proposition 4** (RKHS-SHAP). *Given $f \in \mathcal{H}_k$ and $\nu$, Shapley values $\mathbf{B} \in \mathbb{R}^{d \times n}$ for all $d$ features and all $n$ input $\mathbf{x}$ can be computed as $\mathbf{B} = (Z^\top W Z)^{-1} Z^\top W \hat{\mathbf{V}}$ where $\hat{\mathbf{V}}_{i,:} = \hat{\nu}_{\mathbf{x},S_i}(f)$.*

**Estimating value functions with specific models.** To the best of our knowledge, TreeSHAP [24] was the only machine learning model-specific SV algorithm computing conditional expectations using the properties of the model (tree in this case) directly, rather than relying on some sort of sampling procedure and density estimation. However, it is unclear how to validate the assumptions about feature distribution in TreeSHAP, which are specified as "the distribution generated by the tree", as discussed by Sundararajan and Najmi [41]. In comparison, RKHS-SHAP does not pose assumptions on the underlying feature distribution and computes the corresponding conditional expectations via mean embeddings analytically. However, one should note that each of these model specific algorithm are only designed to explain specific models, therefore it is not informative to compare, e.g. TreeSHAP values with RKHS-SHAP values, as they are explaining different models.

### 3.1 Robustness of RKHS-SHAP

Robustness of interpretability methods is important from both an epistemic and ethical perspective, as discussed in Hancox-Li [17]. On the other hand, Alvarez-Melis and Jaakkola [2] showed empirically that Shapley methods when used with complex non-linear black-box models such as neural networks, yield explanations that vary considerably for some neighbouring inputs, even if the deep network gives similar predictions at those neighbourhoods. In light of this, we analyse the Shapley values obtained from our proposed RKHS-SHAP and show that they are robust. To illustrate this, we first formally define the *Shapley functional*,

**Proposition 5** (Shapley functional). *Given a value functional $\nu$ indexed by input $x$ and coalition $S$, the Shapley functional $\phi_{x,i} : \mathcal{H}_k \to \mathbb{R}$ such that $\phi_{x,i}(f)$ gives the $i^{th}$ Shapley values of $x$ on $f$, has the following Riesz representation in the RKHS: $\phi_{x,i} = \frac{1}{d} \sum_{S \subseteq D \setminus \{i\}} \binom{d-1}{|S|}^{-1} (\nu_{x,S \cup i} - \nu_{x,S})$*

Analogously, we denote $\phi_{x,i}^{(I)}$ and $\phi_{x,i}^{(O)}$ as the *Interventional Shapley functional* (ISF) and *Observational Shapley functional* respectively (OSF). Using the functional formalism, we now show that

given $f \in \mathcal{H}_k$, when $||x - x'||^2 \leq \delta$ for $\delta > 0$, the difference in Shapley values at $x$ and $x'$ will be arbitrarily small for all features i.e. $|\phi_{x,i}(f) - \phi_{x',i}(f)|$ is small $\forall i \in D$. This corresponds to the following,

$$|\phi_{x,i}(f) - \phi_{x',i}(f)|^2 = |\langle f, \phi_{x,i} - \phi_{x',i}\rangle|^2 \leq ||f||^2_{\mathcal{H}_k}||\phi_{x,i} - \phi_{x',i}||^2_{\mathcal{H}_k} \tag{3}$$

where we use Cauchy-Schwarz for the last line. Therefore, for a given $f$ with fix RKHS norm, the key to show robustness lies into bounding the Shapley functionals. In the following theorem, we make two assumptions: (1) the base kernels $k^{(i)}$ for each dimension $i \in D$ are bounded, and (2) the (population) conditional mean embedding functions $\mu_{X_{S^c}|X_S}$ belong to the vector-valued RKHSs $\mathcal{H}_{\Gamma_{X_S}}$ for all coalitions $S \subseteq D$, therefore have finite norms. This assumption is also adopted in Park and Muandet [29, Theorem 4.5].

**Theorem 6** (Bounding Shapley functionals). *Let $k$ be a product kernel with $d$ bounded kernels $|k^{(i)}(x,x)| \leq M$ for all $i \in D$. Denote $M_\mu := \sup_{S \subseteq D} M^{|S|}, M_\Gamma := \sup_{S \subseteq D} ||\mu_{X_{S^c}|X_S}||^2_{\Gamma_{X_S}}$ and $L_\delta = \sup_{S \subseteq D} ||\psi_{x_S} - \psi_{x'_S}||^2_{\mathcal{H}_k}$. Let $\delta > 0$, assume $|x^{(i)} - x^{(i)'}|^2 \leq \delta$ for all features $i \in D$, then differences of the Interventional and Observational Shapley functionals for feature $i$ at observation $x, x'$ can be bounded as $||\phi^{(I)}_{x,i} - \phi^{(I)}_{x',i}||^2_{\mathcal{H}_k} \leq 2M_\mu L_\delta$ and $||\phi^{(O)}_{x,i} - \phi^{(O)}_{x',i}||^2_{\mathcal{H}_k} \leq 4M_\Gamma M_\mu L_\delta$. If $k$ is the RBF kernel with lengthscale $l$, then*

$$||\phi^{(I)}_{x,i} - \phi^{(I)}_{x',i}||^2_{\mathcal{H}_k} \leq 4(1 - \exp(-d\delta/2l^2)), \qquad ||\phi^{(O)}_{x,i} - \phi^{(O)}_{x',i}||^2_{\mathcal{H}_k} \leq 8M_\Gamma(1 - \exp(-d\delta/2l^2))$$

Therefore, as long as $||f||_{\mathcal{H}_k}$ is small, RKHS-SHAP will return robust Shapley values with respect to small perturbations. Notice the Shapley functionals do not depend on $f$ and can be estimated separately purely based on data. We will show in the next section how this key property allows us to use the functional itself to aid in learning of $f$. This enables us to enforce particular structural constraints on $f$ via an additional regularisation term.

## 4 Shapley regularisation

Regularisation is popular in machine learning because it allows inductive bias to be injected to learn functions with specific properties. For example, classical $L_1$ and $L_2$ regularisers are used to control the sparsity and smoothness of model parameters. Manifold regularisation [4], on the other hand, exploits the geometry of the distribution of unlabelled data to improve learning in a semi-supervised setting, whereas Pérez-Suay et al. [31] and Li et al. [22] adopted a kernel dependence regulariser to learn functions for fair regression and fair dimensionality reduction. In the following, we propose a new *Shapley regulariser* based on the Shapley functionals, which allows learning while controlling the level of specific feature's contributions to the model.

**Formulation**   Let $A$ be a specific feature whose contribution we wish to regularise, $f$ the function we wish to learn, and $\phi_{x_i,A}(f)$ the Shapley value of $A$ at a given observation $x_i$. Our goal is to penalise the mean squared magnitude of $\{\phi_{x_i,A}(f)\}^n_{i=1}$ in the ERM framework, which corresponds to $\min_{f \in \mathcal{H}_k} \sum^n_{i=1} \ell(y_i, f(x_i)) + \lambda_f ||f||^2_{\mathcal{H}_k} + \frac{\lambda_S}{n} \sum^n_{i=1} |\phi_{x_i,A}(f)|^2$, where $\ell$ is some loss function and $\lambda_f$ and $\lambda_S$ control the level of regularisations. If we replace the population Shapley functional with the finite sample estimate from Prop. 2, and utilise the Representer theorem, we can rewrite the optimisation in terms of $\boldsymbol{\alpha}$,

**Proposition 7.** *The above optimisation can be rewritten as,* $\min_{\boldsymbol{\alpha} \in \mathbb{R}^n} \sum^n_{i=1} \ell(y_i, \mathbf{K}_{x_i\mathbf{x}}\boldsymbol{\alpha}) + \lambda_f \boldsymbol{\alpha}^\top \mathbf{K}_{\mathbf{xx}}\boldsymbol{\alpha} + \frac{\lambda_S}{n}\boldsymbol{\alpha}^\top \zeta_A \zeta_A^\top \boldsymbol{\alpha}$. *To regularise the Interventional SVs (ISV-REG) of $A$, we set* $\zeta_A = \frac{1}{J}\sum^J_{j=1} \mathcal{K}^{(I)}_{\mathbf{x},S_j \cup A} - \mathcal{K}^{(I)}_{\mathbf{x},S_j}$ *where $S_j$'s are coalitions sampled from $p_{SV}(S) = \frac{1}{d}\binom{d-1}{|S|}^{-1}$. For regularising Observational SVs (OSV-REG), we set* $\zeta_A = \frac{1}{J}\sum^J_{j=1} \mathcal{K}^{(O)}_{\mathbf{x},S_j \cup A} - \mathcal{K}^{(O)}_{\mathbf{x},S_j}$.

In particular, closed form optimal dual weights $\boldsymbol{\alpha} = (\mathbf{K}^2_{\mathbf{xx}} + \lambda_f \mathbf{K}_{\mathbf{xx}} + \frac{\lambda_S}{n}\zeta_A \zeta_A^\top)^{-1} \mathbf{K}_{\mathbf{xx}}\mathbf{y}$ can be recovered when $\ell$ is the squared loss.

**Choice of regularisation.**   Similar to the feature attribution problem, *the choice of regularising against ISVs or OSVs is application dependent* and boils down to whether one wants to take the correlation of $A$ with other features into account or not.

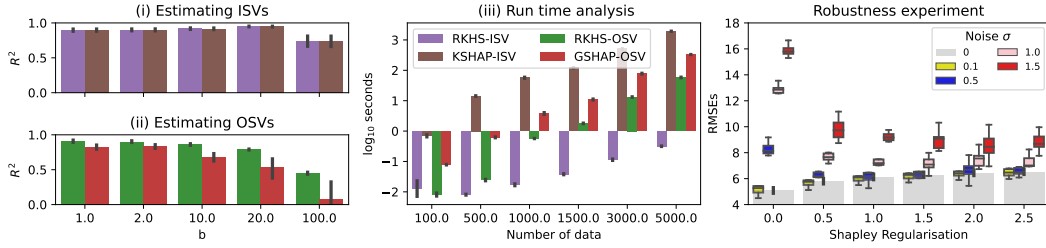

| (a) RKHS-SHAP experiments | (b) ISV-REG experiments |

Figure 2: (a) RKHS-SHAP: Estimation of Shapley values using data from the Banana distribution. Run time analysis in $\log$ scale is also reported. (b) ISV-REG: RMSEs of $f_{\text{reg}}$ on noisy test data at different noise level $\sigma'$. All scores are averaged over 10 runs and 1 sd is reported.

**ISV-REG**    ISV-REG can be used to protect the model when covariate shift of variable $A$ is expected to happen at test time and one wishes to downscale $A$'s contribution during training instead of completely removing this (potentially useful) feature. Such situation may arise if, e.g., a different measurement equipment or process is used for collecting observations of $A$ during test time. ISV is well suited for this problem as dependencies across features will be broken by the covariate shift at test time.

**OSV-REG**    On the other hand, OSV-REG can find its application in fair learning – learning a function that is fair with respect to some sensitive feature $A$. There exist a variety of fairness notions one could consider, such as, e.g. *Statistical Parity*, *Equality of Opportunity* and *Equalised Odds* [7]. In particular, we consider the fairness notion recently explored in the literature [19, 25] that uses Shapley values, which are becoming a bridge between Explainable AI and fairness, given that they can detect biased explanations from biased models. In particular, Jain et al. [19] illustrated that if a model is fair against a sensitive feature $A$, $A$ should have neither a positive nor negative contribution towards the prediction. This corresponds to $A$ having SVs with negligible magnitudes. Simply removing $A$ from the training doesn't make the model fair, as contributions of $A$ might enter the model via correlated features, therefore it is important to take feature correlations into account while regularising. Hence, it is natural to deploy OSV-REG for fair learning.

## 5    Experiments

We demonstrate specific properties of RKHS-SHAP and Shapley regularisers using four synthetic experiments, because these properties are best illustrated under a fully controlled environment. For example, to highlight the merit of distributional-assumption-free value function estimation in RKHS-SHAP, we need groundtruth conditional expectations of value functions for verification, but they are not available in real-world data because we do not observe the true data generating distribution. Nonetheless, as model interpretability is a practical problem, we have also ran several larger scales ($n = 50000, 1.8 \times 10^6$) real-world explanation tasks using RKHS-SHAP and reported our findings in Appendix E for a complete empirical demonstration. All code and implementations are made publicly available [3].

In the first two experiments, we evaluate RKHS-SHAP methods against benchmarks on estimating Interventional and Observational SVs on a Banana-shaped distribution with nonlinear dependencies [34]. The setup allows us to obtain closed-form expressions for the ground truth ISVs and OSVs, yet the conditional distributions among features are challenging to estimate using any standard parametric density estimation methods. We also present a run time analysis to demonstrate empirically that mean embedding based approaches are significantly more efficient than sampling based approaches. Finally, the last two experiments are applications of Shapley regularisers in robust modelling to covariate shifts and fair learning with respect to a sensitive feature.

In the following, we denote RKHS-OSV and RKHS-ISV as the OSV and ISV obtained from RKHS-SHAP. As benchmark, we implement the model agnostic sampling-based algorithm KERNELSHAP from the Python package **shap** [23]. We denote the ISV obtained from KERNELSHAP as KSHAP-ISV. As **shap** does not offer model-agnostic OSV algorithm, we implement the approach from Aas et al. [1] (described in Section 3), where OSVs are estimated using Monte Carlo samples from fitted multivariate Gaussians. We denote this approach as GSHAP-OSV. We fit a kernel ridge regression on each of our experiments. Lengthscales of the kernel are selected using median heuristic [12] and

regularisation parameters are selected using cross-validation. Further implementation details and real world data illustrations are included in Appendix E.

## 5.1  RKHS-SHAP experiments

**Experiment 1: Estimating Shapley values from Banana data.**   We consider the following 2d-Banana distribution $\mathcal{B}(b^{-1}, v)$ from Sejdinovic et al. [34]: Sample $Z \sim N(0, \text{diag}(v, 1))$ and transform the data by setting $X_1 = Z_1$ and $X_2 = b^{-1}(Z_1^2 - v) + Z_2$. Regression labels are obtained from $f_{\text{truth}}(X) = b^{-1}(X_1^2 - v) + X_2$. This formulation allows us to compute the true ISVs and OSVs in closed forms, i.e $\phi_{X,1}^{(I)}(f_{\text{truth}}) = b^{-1}(X_1^2 - v)$, $\phi_{X,2}^{(I)}(f_{\text{truth}}) = X_2$, $\phi_{X,1}^{(O)}(f_{\text{truth}}) = \frac{1}{2}(3b^{-1}(X_1^2 - v) - X_2)$ and $\phi_{X,2}^{(O)}(f_{\text{truth}}) = \frac{1}{2}(3X_2 - b^{-1}(X_1^2 - v))$. In the following we will simulate 3000 data points from $\mathcal{B}(b^{-1}, 10)$ with $b \in [1, 10, 20, 50, 100]$, where smaller values of $b$ correspond to more nonlinearly elongated distributions. We choose $R^2$ as our metric since the true Shapley values for each experiment are scaled according to $b$. Figure 2a(i) and 2a(ii) demonstrate $R^2$ scores of estimated ISVs and OSVs in contrast with groundtruths SVs across different configurations. We see that RKHS-ISV and KSHAP-ISV give exactly the same $R^2$ scores across configurations. This is not surprising as the two methods are mathematically equivalent. While in KSHAP-ISV one averages over evaluated $\{f(x_j')\}$ with $x_j'$ being the imputed data, RKHS-ISV aggregated feature maps of the imputed data first before evaluating at $f$, i.e $\sum_{j=1} f(x_j') = \langle f, \sum_{j=1} \phi(x_j')\rangle_{\mathcal{H}_k} = \langle f, \hat{\mu}_X \rangle_{\mathcal{H}_k}$. However, it is this subtle difference in the order of operations contribute to a significant computational speed difference as we later show in Experiment 2. In the case of estimating OSVs, we see RKHS-OSV is consistently better than GSHAP-OSV at all configurations. This highlights the merit of RKHS-OSV as no density estimation is needed, thus avoiding any potential distribution model misspecification which happens in GSHAP-OSV.

**Experiment 2: Run time analysis.**   In this experiment we sample $n$ data points from $\mathcal{B}(1, 10)$ where $n \in [100, 500, 1000, 1500, 3000, 5000]$ and record the $\log_{10}$ seconds required to complete each algorithm. In practice, as the software documentation of **shap** suggests, one is encouraged to subsample their data before passing to the KERNELSHAP algorithm as the background sampling distribution to avoid slow run time. As this approach speeds up computation at the expense of estimation accuracy since less data is used, for fair comparison with our RKHS-SHAP method which utilises all data, we pass the whole training set to the KERNELSHAP algorithm. Figure 2a(iii) illustrates the run time across methods. We note that the difference in runtime between the two sampling based methods KSHAP-ISV and GSHAP-OSV can be attributed to a different software implementation, but we observe that they are both significantly slower than RKHS-ISV and RKHS-OSV. RKHS-OSV is slower than RKHS-ISV as it involves matrix inversion when computing the empirical CME. In practice, one can trivially subsample data for RKHS-SHAP to achieve further speedups like in the **shap** package, but one can also deploy a range of kernel approximation techniques as discussed in Section 2.2.

## 5.2  Shapley regularisation experiments

For the last two experiments we will simulate 3000 samples from $X \sim N(0, \Sigma)$ with $\text{diag}(\Sigma) = \mathbf{1_5}$ and $\Sigma_{4,5} = \Sigma_{5,4} = 0.9$, 0 otherwise, therefore feature $X_4$ and $X_5$ will be highly correlated. We set our regression labels as $f_{\text{true}}(x) = x^\top \beta$ with $\beta = [1, 2, 3, 4, 10]$, enforcing $X_5$ to be the most influential feature. We use 70% of our data for training and 30% for testing.

**Experiment 3: Protection against covariate shift using ISV-REG.**   For this experiment, we inject extra mean zero Gaussian noise to the most influential feature $X_5$ in the testing set, i.e. $X_5' = X_5 + \sigma' N(0, 1)$ for $\sigma' \in [0, 0.1, 0.5, 1, 1.5]$. We assume that there is an expectation for covariate shift in $X_5$ to occur at test time, due to e.g. a change in the measurement precision – hence, we train our model $f_{\text{reg}}$ using ISV-REG at different regularisation level $\lambda_s$ for $\lambda_s \in [0, 0.5, 1, 1.5, 2, 2.5]$. We then compare RMSEs when no covariate shift is present ($\sigma' = 0$) against RMSEs at different noise levels. The results are shown in Figure 2b. We see that when no regularisation is applied, RMSEs increase rapidly as $\sigma'$ increases, indicating our standard unprotected kernel ridge regressor is sensitive to noises from $X_5'$. As the Shapley regularisation parameter increases, the RMSE of the noiseless case gradually increases too, but RMSEs of the noisy data are much closer to the noiseless case, exhibiting robustness to the covariate shift.

**Experiment 4: Fair learning with OSV-REG** At last, we demonstrate the use of Shapley regulariser to enable fair learning. In this context, as we will see, OSV-REG is the appropriate regulariser. Consider $X_5$ as some sensitive feature which we would like to minimise its contribution during the learning of $f$. Recall $X_4$ is highly correlated to $X_5$ so it contains sensitive information from $X_5$ as well. Figure 3 demonstrates how distributions of ISVs and OSVs of $X_4$ and $X_5$ changes as $\lambda_s$ increases. As regularisation increases, the SVs of $X_5$ becomes more centered at 0, indicating lesser contribution to the model $f_{\text{reg}}$. Similar behavior can be seen from the distribution of $\phi_{X,4}^{(O)}(f_{\text{reg}})$ but not from $\phi_{X,4}^{(I)}$. This illustrates how ISV-REG will propagate unfairness through correlated feature $X_4$ while OSV-REG can take them into account by minimising the contribution of sensitive information during learning.

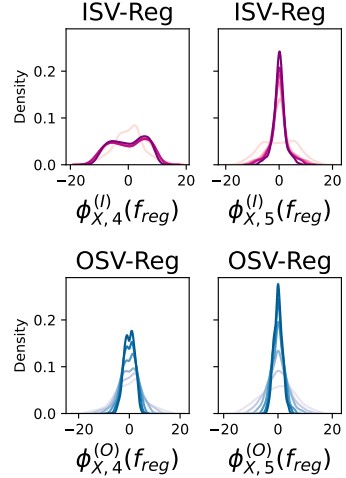

Figure 3: Distributions of SVs of sensitive feature $X_5$ and correlated feature $X_4$ obtained from ISV-REG and OSV-REG at different regularisation parameters. Colour intensity represents the strength of regularisation.

## 6 Conclusion, limitations, and future directions

In this work, we proposed a more accurate and more efficient algorithm to compute Shapley values for kernel methods, termed RKHS-SHAP. We proved that the corresponding local attributions are robust to local perturbations under mild assumptions, a desirable property for consistent model interpretation. Furthermore, we proposed the Shapley regulariser which allows learning while controlling specific feature contribution to the model. We suggested two applications of this regulariser and concluded our work with synthetic experiments demonstrating specific aspects of our contributions. Extensive real-world data explanations are provided in Appendix E.2 for empirical demonstration.

While our methods currently only are applicable to functions arising from kernel methods, a fruitful direction would be to extend the applicability to more general models using the same paradigm. It would also be interesting to extend our formulation to kernel-based hypothesis testing, and for example, to interpret results from two-sample tests.

## Acknowledgements

The authors would like to thank Shahine Bouabid for helpful comments. SLC is supported by the EPSRC and MRC through the OxWaSP CDT programme EP/L016710/1. DS is supported in part by Tencent AI Lab and in part by the Alan Turing Institute (EP/N510129/1).

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
