**RKSH-SHAP: Shapley values for kernel methods supplementary materials**

## A Computational complexity

The gains in speed-up and accuracy in RKHS-SHAP come from estimating $\nu_{\mathbf{x},S}^{(O)}$ using Conditional Mean Embeddings (CMEs). To compare with alternative approaches, it is sufficient to look at the complexity of estimating $\nu_{\mathbf{x},S}^{(O)}(f)$. For RKHS-SHAP this is $\mathcal{O}(Nd^2m) + \mathcal{O}(N^2d^2)$ where $N$ is the number of data, $d$ is the number of Fourier features which could be taken much smaller than $N$ [21] and $m$ is the number of conjugate gradient solver steps. Previous approaches would require some form of density estimation and Monte Carlo sampling, for which there are many methods, so we present a generic decomposition of complexity here: assuming we take $L$ Monte Carlo samples for each $x_{i_S}$ from $p(X_{S^c}|X_S = x_{i_S})$ to estimate $\nu_{\mathbf{x},S}^{(O)}(f)$, we have $\mathcal{O}(L^2N^2) + \mathcal{O}(\text{sampling } NL \text{ data from estimated densities}) + \mathcal{O}(\text{estimating } N \text{ conditional densities})$. It is not clear how to select $L$ nor how fast it should grow with $N$. Aas et al. [1] considered $L = N$ recovering a standard Nadaraya-Waston estimator for their empirical conditional mean estimator. In practice, for nonparametric methods, the computational cost is dominated by density estimation and sampling, both of which are not needed in our approach.

## B Comparison with Frye et al. [13]

As mentioned in the main text, Frye et al. [13]'s approach and ours share a similar regression-like intuition, thus we believe it is important to emphasize the technical difference between the methods.

- **Difference in regression target** Frye's approach regression onto scaler values $f(X)$ for specific $f$ while RKHS-SHAP regresses onto an infinite dimensional feature map instead. Our model is aiming to capture representation of the full conditional distribution via the RKHS embedding, rather than the conditional expectation for a specific $f$.
- **Difference in dependency on** $f$ CME estimation depends on the function space that $f$ belongs to and not on the specific $f$. This subtle but crucial point allows onto apply Shapley functionals as attribution priors during the learning of $f$ itself, in order to regularise it.
- **Difference in hypothesis space:** Frye's approach uses a scaler-valued parametric neural network model, while our approach uses an RKHS-valued non-parametric kernel ridge regression with a ridge penalty to promote smoothness.

## C RKHS-SHAP for non-product kernels

When $k$ is not a product kernel, such as the polynomial kernel and Matérn kernel, we can still proceed with estimating the value function using tools from conditional mean embeddings, and utilise our interpretability pipeline without the need for solving conditional density estimation tasks. To do so, we notice that for any $f \in \mathcal{H}_k$, we have

$$\nu_{x,S}(f) := \mathbb{E}_X[f(X) \mid X_S = x_S] \tag{4}$$

$$= \langle f, \mathbb{E}_X[\psi_X \mid X_S = x_S]\rangle_{\mathcal{H}_k} \tag{5}$$

$$= \langle f, \mu_{X|X_S=x_S}\rangle_{\mathcal{H}_k}. \tag{6}$$

Thus, we can proceed with the following estimator of $\mathbb{E}_X[\psi_X \mid X_S = x_S]$ using the standard conditional mean embedding estimator (with the conditioning variable being the subset of features): Denote $k_S : \mathcal{X}_S \times \mathcal{X}_S \to \mathbb{R}$ as a kernel defined on $\mathcal{X}_S$, where $\mathcal{X}_S$ is the subspace of the instance space of $\mathcal{X}$ according to $S$. Note that in principle, this kernel $k_S$ need not be of the same form as the kernel $k$ defined on the full feature space,

$$\hat{\mu}_{X|X_S=x_S} = \mathbf{K}_{x_S,X_S} \left(\mathbf{K}_{X_S,X_S} + n\lambda I\right)^{-1} \Psi_X^\top. \tag{7}$$

As a result, for $f = \sum_{i=1}^n \alpha_i k(\cdot, x_i)$, the corresponding non-parametric estimator of the value function $\nu_{x,S}(f)$ will be,

$$\bar{\nu}_{x,S}(f) = \mathbf{K}_{x_S,X_S} \left(\mathbf{K}_{X_S,X_S} + n\lambda I\right)^{-1} \mathbf{K}_{X,X}\boldsymbol{\alpha}. \tag{8}$$

where $\boldsymbol{\alpha} = (\alpha_1, \ldots, \alpha_n)^\top$.

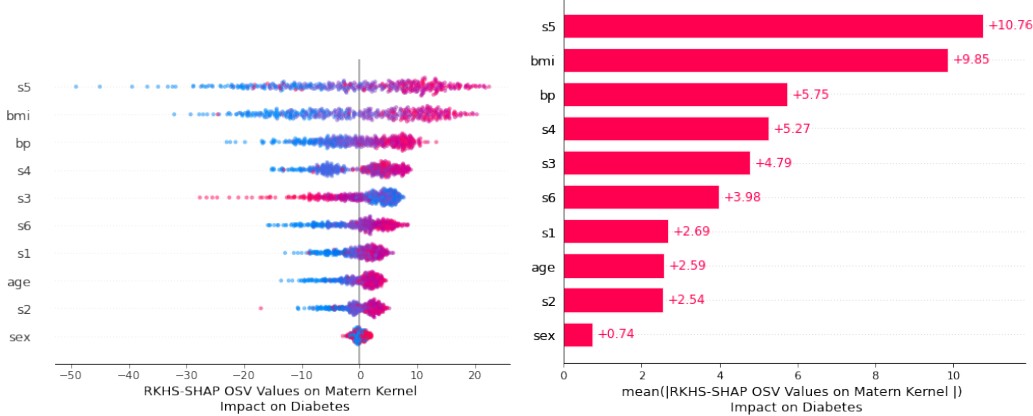

Figure 4: Explaining a Kernel Ridge regression learnt using a Matérn kernel on the Diabetes regression dataset. In comparison to Figure 8, where the KRR uses a Gaussian kernel, we see both models treat feature *s5*, *bp*, and *bmi* as top predictors, but having different emphasises on features *s3* and *s4*.

**Empirical demonstration**   In the following, we will demonstrate the above estimation procedure to explain a kernel ridge regression learnt using Matérn kernel, given by

$$k(x, x') = \frac{1}{\Gamma(v)2^{v-1}} \left( \frac{\sqrt{2v}}{l} \|x - x'\| \right)^v K_v \left( \frac{\sqrt{2v}}{l} \|x - x'\| \right) \tag{9}$$

where $v = 0.5$, $K_v$ is the modified Bessel function of the second kind, and $\Gamma$ is the gamma function. Kernel ridge regression is fitted on the diabetes and housing regression datasets from Appendix E.

Figure 4 and 5 illustrated the explanation results coming from the kernel ridge regression with a Matérn kernel. We refer the reader to Appendix E for a guide to interpret results from the beeswarm and bar plots.

In summary, the product kernel assumption is not required for the benefits of RKHS-SHAP to be brought to bear. Our proposed framework can thus be applied to essentially any kernel appropriate for the problem at hand. It is however, required to specify the form of the said kernel for any subset of features in the case of a non-product kernel, e.g. whether it again takes a Matérn form like the original kernel, or something else. Kernel hyperparameter learning will be more challenging than the product case as well, since e.g. lengthscale parameters typically vary with dimension and one would essentially require one lengthscale per subset of the features we are conditioning on, in contrast to the product case, where one lengthscale per feature dimension suffices. We might incur extra estimation error compared to the product kernel case as well. This is because one must fit the conditional mean embedding for any subset of features individually by regressing to the original RKHS defined on a higher-dimensional space (on all features $d$ rather than on the subset $|S^c|$). As an example, if $d = 100$, in the non-product case we always perform estimation on the space of functions of $100$ arguments, whereas in the product case, if one is conditioning on a $|S| = 99$ dimensional subset, this simplifies to estimation on the space of functions of a scaler argument. Not only is the learning problem harder, the non-product approach has to ignore the fact that the conditioning variable here is simply the subset of features – i.e. standard CME proceeds with regressing from features of $X_S$ to features of $X$, while in the product case it is possible to simply isolate the features we condition on, and set them to the values of interests. As a result, the product kernel assumption allows us to circumvent potential statistical errors, and thus we chose to focus on the product kernel in the main text.

# D   Proofs

## D.1   Proof for Proposition 2.

**Proposition 2** (Riesz representations of value functionals). *Denote $k$ as the product kernel of $D$ bounded kernels $k_d : \mathcal{X}^{(d)} \times \mathcal{X}^{(d)} \to \mathbb{R}$, where $\mathcal{X}^{(d)}$ is the $d^{th}$ feature space. The Riesz representations*

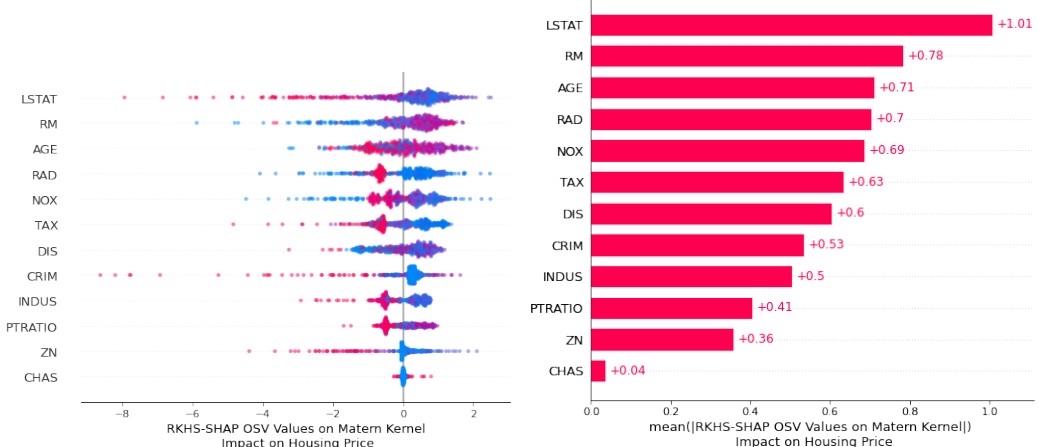

Figure 5: Explaining a Kernel Ridge regression learnt using a Matérn kernel on the House price regression dataset. In comparison to Figure 6, we see that *ZN* is no longer the top predictor. This illustrated that the models emphasised the feature *ZN* very differently.

*of the Interventional value functional and Observational value functional exist and have the following forms in $\mathcal{H}_k$,*

$$\nu_{x,S}^{(I)} = \psi_{x_S} \otimes \mu_{X_{S^c}} \tag{10}$$

$$\nu_{x,S}^{(O)} = \psi_{x_S} \otimes \mu_{X_{S^c}|X_S=x_S} \tag{11}$$

*where $\psi_{x_S} := \bigotimes_{i \in S} \psi_{x^{(i)}}$, $\mu_{X_{S^c}} := \mathbb{E}[\bigotimes_{i \in S^c} \psi_{x^{(i)}}]$ and $\mu_{X_{S^c}|X_S=x_S} := \mathbb{E}[\bigotimes_{i \in S^c} \psi_{x^{(i)}}|X_S = x_S]$.*

*Proof.* Since $\nu_{x,S}^{(I)}$ and $\nu_{x,S}^{(O)}$ are bounded linear functionals on all $f \in \mathcal{H}_k$ with $||f||_{\mathcal{H}_k}$ bounded, Riesz representation theorem [30] tells us there exist $r_{\nu_{x,S}^{(I)}}$ and $r_{\nu_{x,S}^{(O)}}$ living in $\mathcal{H}_k$ such that $\nu_{x,S}^{(I)}(f) = \langle f, r_{\nu_{x,S}^{(I)}} \rangle$ and $\nu_{x,S}^{(O)}(f) = \langle f, r_{\nu_{x,S}^{(O)}} \rangle$. If fact, if we set $r_{\nu_{x,S}^{(I)}}$ to be $\psi_{x_S} \otimes \mu_{X_{S^c}}$ and $r_{\nu_{x,S}^{(O)}}$ to be $\psi_{x_S} \otimes \mu_{X_{S^c}|X_S=x_S}$, then for the former, we have,

$$\langle f, \psi_{x_S} \otimes \mu_{X_{S^c}} \rangle = \mathbb{E}[\langle f, \psi_{x_S} \otimes \psi_{X_{S^c}} \rangle] \tag{12}$$

$$= \mathbb{E}[f(\{x_S, X_{S^c}\})] \tag{13}$$

Similarly,

$$\langle f, \psi_{x_S} \otimes \mu_{X_{S^c}|X_S=x_S} \rangle = \mathbb{E}[\langle f, \psi_{x_S} \otimes \psi_{X_{S^c}} \rangle | X_S = x_S] \tag{14}$$

$$= \mathbb{E}[f(\{x_S, X_{S^c}\})|X_S = x_S] \tag{15}$$

$\square$

## D.2 Proof of Proposition 3.

**Proposition 3.** *Given $\mathbf{x}' \in \mathbb{R}^{n'}$ a vector of instances and $f = \Psi_{\mathbf{x}}\boldsymbol{\alpha}$, the empirical estimates of $\nu_{\mathbf{x}',S}^{(I)}(f)$ and $\nu_{\mathbf{x}',S}^{(O)}(f)$ can be computed as,*

$$\hat{\nu}_{\mathbf{x}',S}^{(I)}(f) = \boldsymbol{\alpha}^\top \mathcal{K}_{\mathbf{x}',S}^{(I)} \qquad \hat{\nu}_{\mathbf{x}',S}^{(O)}(f) = \boldsymbol{\alpha}^\top \mathcal{K}_{\mathbf{x}',S}^{(O)} \tag{16}$$

*where $\mathcal{K}_{\mathbf{x}',S}^{(I)} = \left( \mathbf{K}_{\mathbf{x}_S \mathbf{x}'_S} \odot \frac{1}{n} \operatorname{diag}(\mathbf{K}_{\mathbf{x}_{S^c} \mathbf{x}'_{S^c}}^\top \mathbf{1_n}) \mathbf{1_n} \mathbf{1_n}^\top \right)$ and $\mathcal{K}_{\mathbf{x}',S}^{(O)} = \left( \mathbf{K}_{\mathbf{x}_S \mathbf{x}'_S} \odot \Xi_S \mathbf{K}_{\mathbf{x}_S \mathbf{x}'_S} \right)$, $\mathbf{1_n}$ is the all-one vector with length $n$, $\odot$ the Hadamard product and $\Xi_S = \mathbf{K}_{\mathbf{x}_{S^c} \mathbf{x}_{S^c}} (\mathbf{K}_{\mathbf{x}_S \mathbf{x}_S} + n\eta I)^{-1}$*

*Proof.* Consider $x$ a single observation. Recall $f = \Psi_{\mathbf{x}}\boldsymbol{\alpha}$ and $\Psi_{\mathbf{x}} = [\psi_{x_1}...\psi_{x_n}] = [\psi_{x_{1_S}} \otimes \psi_{x_{1_{S^c}}}...\psi_{x_{n_S}} \otimes \psi_{x_{n_{S^c}}}]$. To compute $\hat{\nu}_{x,S}^{(I)}(f)$, we have:

$$\hat{\nu}_{x,S}^{(I)}(f) = \langle f, \psi_{x_S} \otimes \hat{\mu}_{X_{S^c}} \rangle \tag{17}$$

$$= \langle \Psi_{\mathbf{x}}\boldsymbol{\alpha}, \psi_{x_S} \otimes \frac{1}{n} \sum_{i=1}^{n} \psi_{x_{i_{S^c}}} \rangle \tag{18}$$

$$= \boldsymbol{\alpha}^{\top} \big( \mathbf{K}_{\mathbf{x}_S x_S} \times \frac{1}{n} \mathbf{K}_{\mathbf{x}_{S^c} x_{S^c}}^{\top} \mathbf{1}_n \big) \tag{19}$$

Similarly, for $\nu_{x,S}^{(O)}(f)$,

$$\hat{\nu}_{x,S}^{(O)}(f) = \langle f, \psi_{x_S} \otimes \hat{\mu}_{X_{S^c}|X_S=x_S} \rangle \tag{20}$$

$$= \langle \Psi_{\mathbf{x}}\boldsymbol{\alpha}, \psi_{x_S} \otimes \Psi_{\mathbf{x}_{S^c}} (\mathbf{K}_{\mathbf{x}_S \mathbf{x}_S} + \eta I)^{-1} \mathbf{K}_{\mathbf{x}_S x_S} \rangle \tag{21}$$

$$= \boldsymbol{\alpha}^{\top} \big( \mathbf{K}_{\mathbf{x}_S x_S} \odot \mathbf{K}_{\mathbf{x}_{S^c} \mathbf{x}_{S^c}} (\mathbf{K}_{\mathbf{x}_S \mathbf{x}_S} + n\eta I)^{-1} \mathbf{K}_{\mathbf{x}_S x_S} \big) \tag{22}$$

Extension to a vector of instance $\mathbf{x}'$ is then straight forward. $\qquad\square$

### D.3 Proof of Proposition 4.

**Proposition 4** (RKHS-SHAP). *Given $f \in \mathcal{H}_k$ and a value functional $\nu$, Shapley values for all $d$ features and all input $\mathbf{x}$ can be computed as follows:*

$$\mathbf{B} = (Z^{\top} W Z)^{-1} Z^{\top} W \hat{\mathbf{V}} \tag{23}$$

*where $\hat{\mathbf{V}}_{i,:} = \langle f, \hat{\nu}_{\mathbf{x},S_i} \rangle$.*

**Proof**   Since we now have a compact way to estimate the conditional estimations for a vector of observations using mean embeddings, we can restate the KernelSHAP objective, which essentially is a weighted least regression, into a multi-output weighted least square formulation.

### D.4 Proof of Proposition 5.

**Proposition 5** (Shapley functional). *Given a value functional $\nu$ indexed by input $x$ and coalition $S$, the Shapley functional $\phi_{x,i} : \mathcal{H}_k \to \mathbb{R}$ such that $\phi_{x,i}(f)$ is the $i^{th}$ Shapley values for model $f$ on input $x$, has the following Reisz representation in the RKHS,*

$$\phi_{x,i} = \frac{1}{d} \sum_{S \subseteq D \setminus \{i\}} \binom{d-1}{|S|}^{-1} \big( \nu_{x,S \cup i} - \nu_{x,S} \big) \tag{24}$$

**Proof**   Since the Shapley functional is a linear combination of bounded linear functionals (value functionals), it admits a Riesz representer in the RKHS.

### D.5 Proof of Theorem 6.

**Theorem 6** (Bounding Shapley functionals). *Let $k$ be a product kernel with $d$ bounded kernels $|k^{(i)}(x,x)| \leq M$ for all $i \in D$. Denote $M_{\mu} := \sup_{S \subseteq D} M^{|S|}$, $M_{\Gamma} := \sup_{S \subseteq D} ||\mu_{X_{S^c}|X_S}||_{\Gamma_{X_S}}^2$ and $L_{\delta} = \sup_{S \subseteq D} ||\psi_{x_S} - \psi_{x'_S}||_{\mathcal{H}_k}^2$. Let $\delta > 0$, assume $|x^{(i)} - x^{(i)'}|^2 \leq \delta$ for all features $i \in D$, then differences of the Interventional and Observational Shapley functionals for feature $i$ at observation $x, x'$ can be bounded as $||\phi_{x,i}^{(I)} - \phi_{x',i}^{(I)}||_{\mathcal{H}_k}^2 \leq 2M_{\mu}L_{\delta}$ and $||\phi_{x,i}^{(O)} - \phi_{x',i}^{(O)}||_{\mathcal{H}_k}^2 \leq 4M_{\Gamma}M_{\mu}L_{\delta}$. If $k$ is the RBF kernel with lengthscale $l$, then*

$$||\phi_{x,i}^{(I)} - \phi_{x',i}^{(I)}||_{\mathcal{H}_k}^2 \leq 4\Big(1 - \exp\Big(\frac{-d\delta}{2l^2}\Big)\Big) \tag{25}$$

$$||\phi_{x,i}^{(O)} - \phi_{x',i}^{(O)}||_{\mathcal{H}_k}^2 \leq 8M_{\Gamma}\Big(1 - \exp\Big(\frac{-d\delta}{2l^2}\Big)\Big) \tag{26}$$

**Proof** To prove that Shapley functionals between two observations $x$ and $x'$ are $\delta$ close when the two points are closed, we proceed as follows: (1) We show that when one pick the usual product RBF kernel, we can bound the distance of the feature maps as a function of $\delta$. (2) We then upper bound the value functionals and show that this bound can be relaxed so that it is independent with the choice of coalition. (3) Since Shapley values is an expectation of differences of value functions, by devising a coalition independent bound for the difference in value functionals, the expectation disappears in our bound.

**Proposition A.1** (Bounding feature maps). *For the simplest 1 dimensional case with $|x - x'|^2 \leq \delta$, if we pick $k$ the standard RBF kernel with lengthscale $l$, we have,*

$$||\psi_x - \psi_{x'}||^2_{\mathcal{H}_k} \leq 2 - 2 \exp\left(-\frac{\delta}{2l^2}\right) \tag{27}$$

*When we pick $x, x' \in \mathbb{R}^d$ and with a product RBF kernel i.e $k(x, x') = \prod_{j=1}^d k^j(x^{(j)}, x'^{(j)})$, where $k^{(j)}$ themselves RBF kernels. For simplicity, we assume they all share the same lengthscale $l$. If $|x^{(j)} - x'^{(j)}| \leq \delta$ for all $j \in D$, then we can bound the difference in feature maps as follows,*

$$||\psi_x - \psi_{x'}||^2_{\mathcal{H}_k} \leq 2 - 2 \exp\left(-\frac{d\delta}{2l^2}\right) \tag{28}$$

*Proof.* Since $||\psi_x - \psi_{x'}||^2_{\mathcal{H}_k} = k(x, x) + k(x, x') - 2k(x, x')$. Therefore the first 2 terms are 1 and we can bound the last term since,

$$k(x, x') = \exp\left(-\frac{|x - x'|^2}{2l^2}\right) \geq \exp\left(-\frac{\delta}{2l^2}\right) \tag{29}$$

Multiply this lower bound $d$ times to obtain the bound for the $d$ dimensional case. $\square$

Proposition A.1 tells us how the distance in feature maps $||k_x - k_{x'}||_{\mathcal{H}_k}$ can be expressed by the distance between $x$ and $x'$ in the RBF kernel. Different bounds can be derived for different kernels and we only show the special RBF case for illustration purpose.

Now we shall prove a bound for the value functionals. We shall first proceed with the interventional case and move on to observational afterwards.

**Proposition A.2** (Bounding Interventional value functionals). *For a fix coalition $S$, denote $D_S^{(I)} = ||\nu_{x,S}^{(I)} - \nu_{x',S}^{(I)}||^2_{\mathcal{H}_k}$. Then $D_S^{(I)} \leq ||\psi_{x_S} - \psi_{x'_S}||^2_{\mathcal{H}_{k_S}}||\mu_{X_{S^c}}||^2_{\mathcal{H}_{k_{S^c}}}$. Let $L_\delta := \sup_{S \subseteq D} ||\psi_{x_S} - \psi_{x'_S}||^2_{\mathcal{H}_{k_S}}$ and assume kernels are all bounded per dimension by $M$, i.e $k^{(j)}(x, x') \leq M$ for all $j \in D$. Denote $M_\mu := \sup_{S \subseteq D} M^{|S|}$. Then the bound can be further loosen up,*

$$D_S^{(I)} \leq M_\mu L_\delta \tag{30}$$

*Proof.*

$$D_S^{(I)} = ||\nu_{x,S}^{(I)} - \nu_{x',S}^{(I)}||^2_{\mathcal{H}_k} \tag{31}$$

$$= ||\psi_{x_S} \otimes \mu_{X_{S^c}} - \psi_{x'_S} \otimes \mu_{X_{S^c}}||^2_{\mathcal{H}_k} \tag{32}$$

$$\leq ||\psi_{x_S} - \psi_{x'_S}||^2_{\mathcal{H}_{k_S}}||\mu_{X_{S^c}}||^2_{\mathcal{H}_{k_{S^c}}} \tag{33}$$

Note that $||\mu_{X_{S^c}}||^2 = ||\mathbb{E}[k(X_{S^c}, X'_{S^c})]||^2 \leq M^{2|S^c|}$, therefore,

$$\leq L_\delta M_\mu \tag{34}$$

$\square$

Before we proof the main theorem, we will illustrate the following bounds for conditional mean embeddings, which will be used to bound the observational shapley functionals.

**Proposition A.3** (Bounding conditional mean embeddings). *If we take on the vector-valued function perspective of conditional mean embeddings as in Grünewälder et al. [16], then we could assume, in general for random variable $Y$ and $X$, there exits a function $\mu_{Y|X} \in \mathcal{H}_{\Gamma_x}$ where $\Gamma_x : \mathcal{X} \times \mathcal{X} \mapsto \mathcal{L}(\mathcal{H}_\ell)$ with $\mathcal{L}(\mathcal{H}_\ell)$ being the space of self-adjoint operators from the RKHS $\mathcal{H}_\ell$ to itself, is the*

*vector-valued kernel* $\Gamma_x(x, x') = k(x, x')\mathbf{1}$, *such that* $\mu_{Y|X}(x) = \mu_{Y|X=x}$. *If we assume such function exists, then by definition of vector-valued RKHSs as in Park and Muandet [29],* $||\mu_{Y|X}||_{\mathcal{H}_{\Gamma_x}}$ *has finite norm. Therefore the following is defined if the base kernel* $k$ *is bounded,*

$$||\mu_{Y|X=x}||_{\mathcal{H}_\ell} \le ||\mu_{Y|X}||_{\mathcal{H}_{\Gamma_x}}||\psi_x||_{\mathcal{H}_k} \tag{35}$$

*and correspondingly,*

$$||\mu_{Y|X=x} - \mu_{Y|X=x'}||_{\mathcal{H}_\ell} \le ||\mu_{Y|X}||_{\mathcal{H}_{\Gamma_x}}||\psi_x - \psi_{x'}|| \tag{36}$$

*Proof.* For the first claim, using the result from Micchelli and Pontil [26, Prop.1 ], we have,

$$||\mu_{Y|X}(x)||_{\mathcal{H}_\ell} \le ||\mu_{Y|X}||_{\mathcal{H}_{\Gamma_x}}||\Gamma_x(x, x)||_{op}^{\frac{1}{2}} \tag{37}$$

however, we have,

$$||\Gamma_x(x, x)||_{op} = \sup_{g \in \mathcal{H}_\ell} \frac{||k(x, x)g||_{\mathcal{H}_\ell}}{||g||_{\mathcal{H}_\ell}} = |k(x, x)| = ||\psi_x||^2 \tag{38}$$

For the second part, we start with,

$$||\mu_{Y|X=x} - \mu_{Y|X=x'}||_{\mathcal{H}_\ell} \le ||\mu_{Y|X}||_{\mathcal{H}_{\Gamma_x}}||\Gamma_x(\cdot, x) - \Gamma_x(\cdot, x')||_{op} \tag{39}$$

Using the result from Micchelli and Pontil [26, Prop.1] again, we have

$$||\Gamma_x(\cdot, x) - \Gamma_x(\cdot, x')||_{op} = ||\big(\Gamma_x(\cdot, x) - \Gamma_x(\cdot, x')\big)^*\big(\Gamma_x(\cdot, x) - \Gamma_x(\cdot, x')\big)||_{op}^{\frac{1}{2}} \tag{40}$$

where the $*$ denotes the adjoint of the operator,

$$= ||\Gamma_x(\cdot, x)^*\Gamma_x(\cdot, x) - 2\Gamma_x(\cdot, x)^*\Gamma_x(\cdot, x') + \Gamma_x(\cdot, x')^*\Gamma_x(\cdot, x')||_{op}^{\frac{1}{2}} \tag{41}$$

$$= ||\Gamma_x(x, x) - 2\Gamma_x(x, x') + \Gamma_x(x', x')||_{op}^{\frac{1}{2}} \tag{42}$$

$$= ||\big(k(x, x) - 2k(x, x') + k(x', x')\big)\mathbf{1}||_{op}^{\frac{1}{2}} \tag{43}$$

$$= ||\psi_x - \psi_{x'}||_{\mathcal{H}_k} \tag{44}$$

therefore we have as a result,

$$||\mu_{Y|X=x} - \mu_{Y|X=x'}||_{\mathcal{H}_\ell} \le ||\mu_{Y|X}||_{\mathcal{H}_{\Gamma_x}}||\psi_x - \psi_{x'}||_{\mathcal{H}_k} \tag{45}$$

$\square$

**Proposition A.4** (Bounding Observational value functionals via vector-valued function perspective of CME). *For a fix coalition* $S$, *denote* $D_S^{(O)} = ||\nu_{x,S}^{(I)} - \nu_{x',S}^{(I)}||_{\mathcal{H}_k}^2$. *Then* $D_S^{(O)} \le ||\psi_{x_S} - \psi_{x'_S}||_{\mathcal{H}_{k_S}}^2 ||\mu_{X_{S^c}|X_S}||_{\mathcal{H}_{\Gamma_{X_S}}}^2 \big(||\psi_{x_S}||_{\mathcal{H}_{k_S}}^2 + ||\psi_{x'_S}||_{\mathcal{H}_{k_S}}^2\big)$, *where* $\mathcal{H}_{\Gamma_{X_S}}$ *is the* $\mathcal{H}_{k_{S^c}}$-*valued RKHS. If we denote* $L_\delta = \sup_{S \subseteq D} ||\psi_{x_S} - \psi_{x'_S}||_{\mathcal{H}_{k_S}}^2$ *and* $M_\mu = \sup_{S \subseteq D} M^{|S|}$ *and* $M_\Gamma = \sup_{S \subseteq D} ||\mu_{X_{S^c}|X_S}||_{\mathcal{H}_{\Gamma_{X_S}}}^2$. *Then* $D_S^{(O)} \le 2M_\Gamma M_\mu L_\delta$ *for all coalition S.*

*Proof.*

$$D_S^{(O)} = ||\nu_{x,S}^{(O)} - \nu_{x',S}^{(O)}||_{\mathcal{H}_k}^2 \tag{46}$$

$$= ||\psi_{x_S} \otimes \mu_{X_{S^c}|X_S=x_S} - \psi_{x'_S} \otimes \mu_{X_{S^c}|X_S=x'_S}||_{\mathcal{H}_k}^2 \tag{47}$$

$$= ||\psi_{x_S} \otimes \mu_{X_{S^c}|X_S=x_S} - \psi_{x'_S} \otimes \mu_{X_{S^c}|X_S=x_S} + \psi_{x'_S} \otimes \mu_{X_{S^c}|X_S=x_S} - \psi_{x'_S} \otimes \mu_{X_{S^c}|X_S=x'_S}||_{\mathcal{H}_k}^2 \tag{48}$$

$$\le ||\psi_{x_S} - \psi_{x'_S}||_{\mathcal{H}_{k_S}}^2 ||\mu_{X_{S^c}|X_S=x_s}||_{\mathcal{H}_{k_{S^c}}}^2 + ||\psi_{x'_S}||_{\mathcal{H}_{k_S}}^2 ||\mu_{X_{S^c}|X_S=x_S} - \mu_{X_{S^c}|X_S=x'_S}||_{\mathcal{H}_{k_{S^c}}}^2 \tag{49}$$

$$\le ||\psi_{x_S} - \psi_{x'_S}||_{\mathcal{H}_{k_S}}^2 ||\mu_{X_{S^c}|X_S}||_{\mathcal{H}_{\Gamma_{X_S}}}^2 ||\psi_{x_S}||_{\mathcal{H}_{k_S}}^2 + ||\psi_{x'_S}||_{\mathcal{H}_{k_S}}^2 ||\mu_{X_{S^c}|X_S}||_{\mathcal{H}_{\Gamma_{X_S}}}^2 ||\psi_{x_S} - \psi_{x'_S}||_{\mathcal{H}_{k_S}}^2 \tag{50}$$

$$= ||\psi_{x_S} - \psi_{x'_S}||_{\mathcal{H}_{k_S}}^2 ||\mu_{X_{S^c}|X_S}||_{\mathcal{H}_{\Gamma_{X_S}}}^2 \big(||\psi_{x_S}||_{\mathcal{H}_{k_S}}^2 + ||\psi_{x'_S}||_{\mathcal{H}_{k_S}}^2\big) \tag{51}$$

$$\le 2M_\Gamma M_\mu L_\delta \tag{52}$$

Finally, we note that,

$$||\phi_{x,i} - \phi_{x',i}||^2_{\mathcal{H}_k} = ||\frac{1}{d} \sum_{S \subseteq D \setminus \{i\}} \binom{d-1}{|S|}^{-1} \nu_{x,S \cup i} - \nu_{x,S} - (\nu_{x',S \cup i} - \nu_{x',S})||^2_{\mathcal{H}_k} \qquad (53)$$

$$\leq \frac{1}{d} \sum_{S \subseteq D \setminus \{i\}} \binom{d-1}{|S|}^{-1} D_S + D_{S \cup i} \qquad (54)$$

$$= \mathbb{E}_S[D_S + D_{S \cup i}] \qquad (55)$$

Since we have proven bounds for $D_S^{(O)}$ and $D_S^{(I)}$ that is coalition independent, we can directly substitute the bound inside the expectation. Therefore

$$||\phi_{x,i}^{(I)} - \phi_{x',i}^{(I)}||^2_{\mathcal{H}_k} \leq 2L_\delta M_\mu \qquad (56)$$

$$||\phi_{x,i}^{(O)} - \phi_{x',i}^{(O)}||^2_{\mathcal{H}_k} \leq 4M_\Gamma L_\delta M_\mu \qquad (57)$$

In the case when we pick $k$ as a product RBF kernel, we have $L_\delta = 2 - 2\exp\left(\frac{d\delta}{2l^2}\right)$ and $M_\mu = 1$, therefore,

$$||\phi_{x,i}^{(I)} - \phi_{x',i}^{(I)}||^2_{\mathcal{H}_k} \leq 4\left(1 - \exp\left(\frac{-d\delta}{2l^2}\right)\right) \qquad (58)$$

$$||\phi_{x,i}^{(O)} - \phi_{x',i}^{(O)}||^2_{\mathcal{H}_k} \leq 8M_\Gamma\left(1 - \exp\left(\frac{-d\delta}{2l^2}\right)\right) \qquad (59)$$

$\square$

**Proposition 7.** *The above optimisation can be rewritten as,* $\min_{\alpha \in \mathbb{R}^n} \sum_{i=1}^n \ell(y_i, \mathbf{K}_{x_i \mathbf{x}} \alpha) + \lambda_f \alpha^\top \mathbf{K}_{\mathbf{xx}} \alpha + \frac{\lambda_S}{n} \alpha^\top \zeta_A \zeta_A^\top \alpha$. *To regularise the Interventional SVs (ISV-REG) of A, we set* $\zeta_A = \frac{1}{J} \sum_{j=1}^J \mathcal{K}_{\mathbf{x},S_j \cup A}^{(I)} - \mathcal{K}_{\mathbf{x},S_j}^{(I)}$ *where $S_j$'s are coalitions sampled from* $p_{SV}(S) = \frac{1}{d}\binom{d-1}{|S|}^{-1}$. *For regularising Observational SVs (OSV-REG), we set* $\zeta_A = \frac{1}{J} \sum_{j=1}^J \mathcal{K}_{\mathbf{x},S_j \cup A}^{(O)} - \mathcal{K}_{\mathbf{x},S_j}^{(O)}$.

*Sketch proof.* To express

$$\min_{f \in \mathcal{H}_k} \sum_{i=1}^n \ell(y_i, f(x_i)) + \lambda_f ||f||^2_{\mathcal{H}_k} + \frac{\lambda_S}{n} \sum_{i=1}^n |\phi_{x_i,A}(f)|^2$$

as

$$\min_{\alpha \in \mathbb{R}^n} \sum_{i=1}^n \ell(y_i, \mathbf{K}_{x_i \mathbf{x}} \alpha) + \lambda_f \alpha^\top \mathbf{K}_{\mathbf{xx}} \alpha + \frac{\lambda_S}{n} \alpha^\top \zeta_A \zeta_A^\top \alpha,$$

it suffices to show that $\frac{\lambda_S}{n} \sum_{i=1}^n |\phi_{x_i,A}(f)|^2 = \frac{\lambda_S}{n} \alpha^\top \zeta_A \zeta_A^\top \alpha$. However, note that

$$\sum_{i=1}^n |\phi_{x_i,A}(f)|^2 = \phi_{\mathbf{x},A}(f)^\top \phi_{\mathbf{x},A}(f) \qquad (60)$$

$$= f^\top \phi_{\mathbf{x},A} \phi_{\mathbf{x},A}^\top f \qquad (61)$$

Now we can estimate the Shapley functional $\phi_{\mathbf{x},A}$ defined in Proposition 5, by applying the finite sample estimator of the value functions from Proposition 2, we can compute the finite sample estimate of $\phi_{\mathbf{x},A}^\top f$ as $\zeta_A^\top \alpha$. $\square$

# E  Further experiment details

## E.1  Banana Distribution $\mathcal{B}(b^{-1}, v)$

Recall the Banana distribution $\mathcal{B}(b^{-1}, v)$ is defined as follows: Let $Z \sim N(0, \mathrm{diag}(v, 1))$ and set $X_1 = Z_1$ and $X_2 = b^{-1}(Z_1^2 - v) + Z_2$. We define $f(x) = b^{-1}(x_1^2 - v) + x_2$. Now then we have,

$$\mathbb{E}[f(X)] = 0 \tag{62}$$

$$\mathbb{E}[f(X)|X_1 = x_1] = 2b^{-1}(x_1^2 - v) \tag{63}$$

$$\mathbb{E}[f(X)|X_2 = x_2] = 2x_2 \tag{64}$$

$$\mathbb{E}[f(X)|do(X_1) = x_1] = b^{-1}(x_1^2 - v) \tag{65}$$

$$\mathbb{E}[f(X)|do(X_2) = x_2] = x_2 \tag{66}$$

This corresponds to the following Observational Shapley values,

$$
\begin{aligned}
\phi_{x,1}^{(O)}(f) &= \frac{1}{2}\left[ \binom{1}{0}^{-1} (\mathbb{E}f(\mathbf{X}|X_1 = x_1) - \mathbb{E}f(\mathbf{X})) + \binom{1}{1}^{-1} (\mathbb{E}f(\mathbf{X}|X_1 = x_1, X_2 = x_2) - \mathbb{E}f(\mathbf{X}|X_2 = x_2)) \right] \\
&= \frac{1}{2}\left( 3b^{-1}(x_1^2 - v) - x_2 \right). \\
\phi_{x,2}^{(O)}(f) &= \frac{1}{2}\left[ \binom{1}{0}^{-1} (\mathbb{E}f(\mathbf{X}|X_2 = x_2) - \mathbb{E}f(\mathbf{X})) + \binom{1}{1}^{-1} (\mathbb{E}f(\mathbf{X}|X_1 = x_1, X_2 = x_2) - \mathbb{E}f(\mathbf{X}|X_1 = x_1)) \right] \\
&= \frac{1}{2}\left( 3x_2 - b^{-1}(x_1^2 - v) \right)
\end{aligned}
$$

Similarly, for Interventional Shapley values we have,

$$\phi_{x,1}^{(I)}(f) = b^{-1}(x_1^2 - v)$$

$$\phi_{x,2}^{(I)}(f) = x_2$$

## E.2  RKHS-SHAP on real-world examples

We demonstrate the result of running RKHS-SHAP on 6 real-world datasets and showcase their RKHS-SHAP Observational Shapley values in Beeswarm summary plots. Interventional SVs are omitted because we have shown in the main text that running KernelSHAP-ISV and RKHS-SHAP-ISV gives you the same SVs, and they only differ in computational run time.

These results are not included in the main text because **we do not observe the actual data distribution, thus there are no groundtruth observational SVs that our algorithm can be compared to measure and verify how well it is performing**. In the following, all models are fitted with the Gaussian kernel. We first fit a Kernel Ridge Regression or Kernel Logistic Regression to learn the function $f$, and apply RKHS-SHAP to $f$ to recover the corresponding observational Shapley values.

We present our results using Beeswarm plot and bar plot. According to the **shap** package, the beeswarm plot is designed to display an information-dense summary of how the top features in the dataset impact the model's output. Each instance the given explanation is represented by a single dot on each feature row. The x position of the dot is determined by the RKHS-SHAP value of that feature, and dots "pile up" along each feature row to show density. Colour is used to display the original value of a feature, which is scaled with red indicating high, and blue indicating low values. On the other hand, the bar plot shows the mean absolute value of the Shapley values per feature, thus providing some global summary based on recovered local importances.

We summarise our real-world explanation tasks in table 1.

**Boston Housing**  The Boston house price dataset[4] contains 506 instances and 12 numerical features. Below is the description of its features:

---

[4]https://archive.ics.uci.edu/ml/machine-learning-databases/housing/

Table 1: Real-world explanation tasks

| Dataset | $n_{instances}$ | $n_{features}$ | Downstream task |
|---|---|---|---|
| Boston Housing | 506 | 12 | Predict Boston House Price (Regression) |
| Diabetes Progression | 442 | 10 | Predict diabetes progression (Regression) |
| Diabetes for Pima Indian Heritage | 768 | 8 | Predict whether a patient has diabetes (Classification) |
| Breast Cancer | 569 | 30 | Predict whether a patient might have breast cancer or not (Classification) |
| Census Income | 48,842 | 14 | Predict whether an individual is making over $50k a year (Classification) |
| League of Legends Win Prediction | 1,800,000 | 71 | Predict the winning probability of a player (Classification) |

| | |
|---|---|
| CRIM | per capita crime rate by town |
| ZN | proportion of residential land zoned for lots over 25k sq.ft |
| INDUS | proportion of non-retail business acres per town |
| CHAS | Charles River dummy variable (= 1 if tract bounds river; 0 otherwise) |
| NOX | nitric oxides concentration (parts per 10 million) |
| RM | average number of rooms per dwelling |
| AGE | proportion of owner-occupied units built prior to 1940 |
| DIS | weighted distances to five Boston employment centres |
| RAD | index of accessibility to radial highways |
| TAX | full-value property-tax rate per $10,000 |
| PTRATIO | pupil-teacher ratio by town |
| LSTAT | % lower status of the population |
| MEDV | Median value of owner-occupied homes in $1000's |

We fit a Kernel Ridge Regression to predict the Boston house price. The results are shown in Fig. 6. We see that RKHS-SHAP does capture several intuitive explanations, e.g. Higher crime rate (red dots in feature CRIM) corresponds to negative impact on the house price. We also recover explanations such as lower percentage of lower status of the population (LSTAT) will increase the house price.

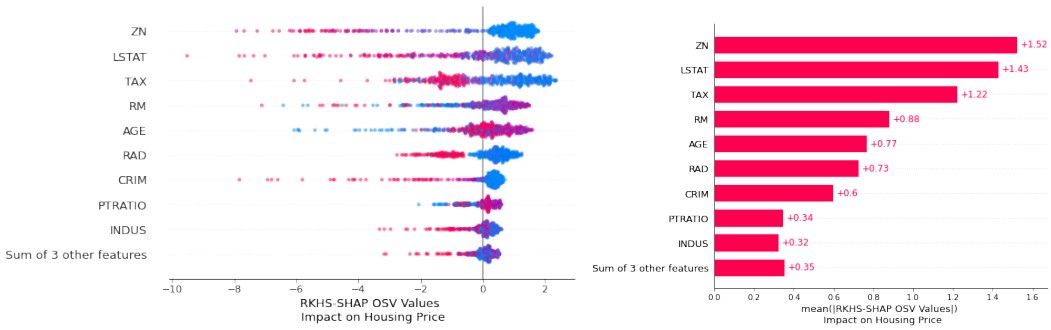

Figure 6: Beeswarm and bar plot for the housing dataset.

We can examine specific houses and interpret why the kernel ridge regression predicts their corresponding house prices as well. See Fig 7.

**Diabetes progression regression**  Next we apply RKHS-SHAP to the diabetes[5] dataset with 442 samples and 10 features. The machine learning task is to model the disease progression of patients as

---

[5]https://www4.stat.ncsu.edu/ boos/var.select/diabetes.html

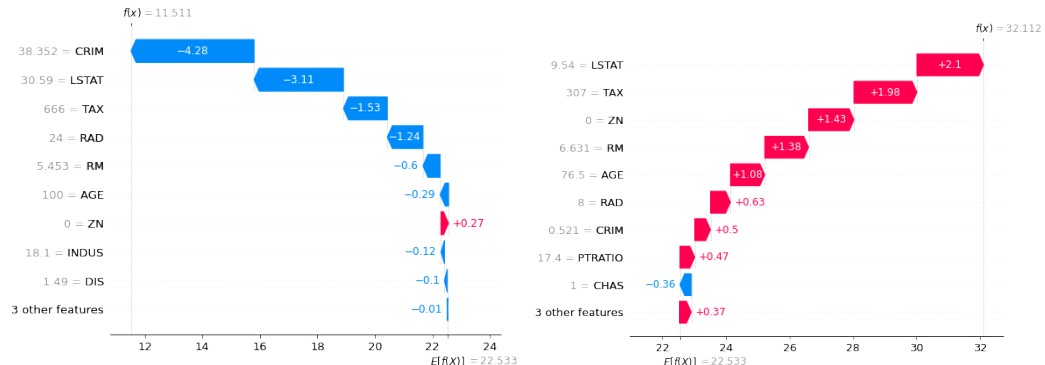

Figure 7: (left) The algorithm believes having a high crime rate is the major reason for its low house price. (Right) Having a high LSTAT increased the house price.

a regression problem. We fit a kernel ridge regression for that. Figure 8 records the results. Feature $s1$ to $s6$ are blood serum measurements. We note that $bmi$ is one of the most influential feature, which follows our intuition that higher value of $bmi$ (red clusters in the bmi row) should be a strongly predictive variable to diabetes.

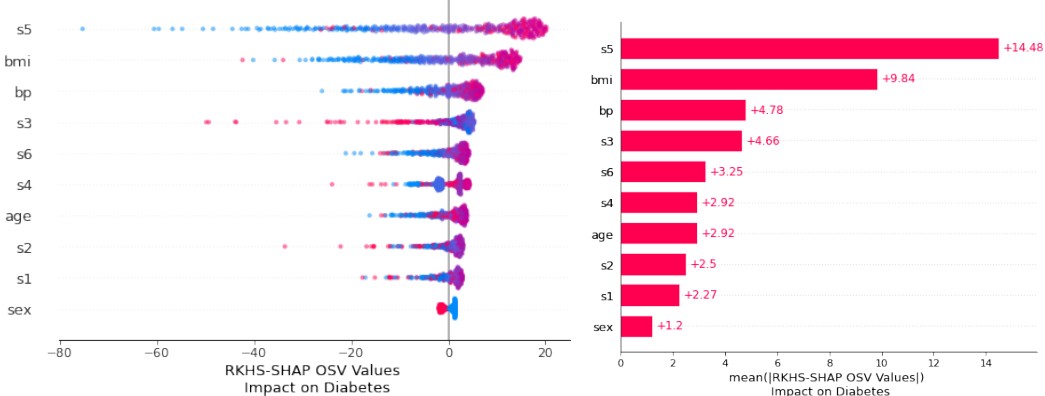

Figure 8: Beeswarm and barplot of the RKHS-SHAP values on the Diabetes dataset

**Diabetes for Pima Indian heritage**   Here we consider another dataset of diabetes study for Pima Indian heritage women aged 21 over. The data set is collected from here[6]. There are 768 samples with 8 features. The goal is to predict whether a patient has diabetes and fit a kernel logistic regression.

Figure 9 demonstrated how RKHS-SHAP explains the kernel logistic regression. The top predictor, "Glucose", which measures the plasma glucose concentration 2 hours in an oral glucose tolerance test, aligns with the intuition that it should be strongly predictive to whether a person is diabetic. Also, high BMI leading to someone more likely to be diabetic is also reflected from RKHS-SHAP values.

**Breast Cancer Classification**   Next, we apply RKHS-SHAP to the breast cancer wisconsin dataset[7] to interpret the kernel logistic regression we have fitted to predict whether a patient might have breast cancer given their attributes. Features are computed from a digitized image of a fine needle aspirate (FNA) of a breast mass. They describe characteristics of the cell nuclei present in the medical image. There are 569 data and 30 features. When running RKHS-SHAP, we did not use all $2^{30}$ coalitions but subsampled 10000 coalitions instead. Convergence analysis of such an approach is studied extensively by [8], where they empirically show that the algorithm will converge in $\mathcal{O}(n)$. Results are shown in Figure 10. We can see that features such as "worst radius", "worst concave

---

[6]https://www.kaggle.com/datasets/mathchi/diabetes-data-set?resource=download
[7]https://goo.gl/U2Uwz2

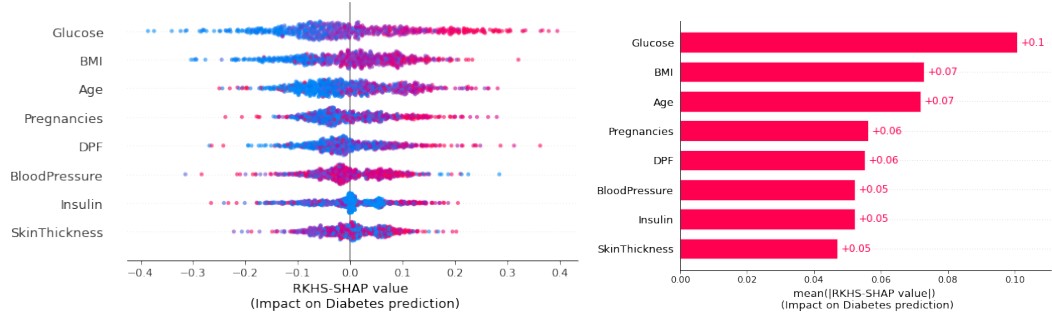

Figure 9: Beeswarm and barplot of the RKHS-SHAP values on the Diabetes for pima indian heritage dataset

points", "worst perimeter" that describes the cell nuclei present in the breast mass, are most predictive to whether a patient has cancer or not.

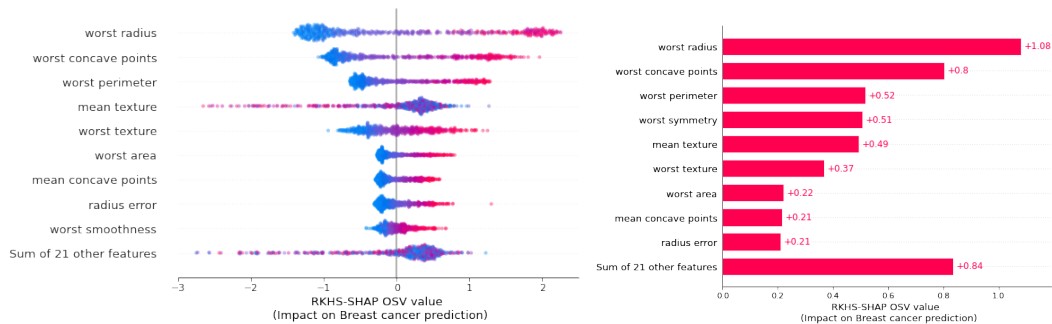

Figure 10: Beeswarm and barplot for the breast cancer prediction problem

**Census Income dataset** In the following, we will explain the kernel logistic regression deployed to predict the probability of an individual making over $ 50K a year in annual income using the standard UCI adult income dataset. There are 48,842 number of instances and 14 attributes. We see

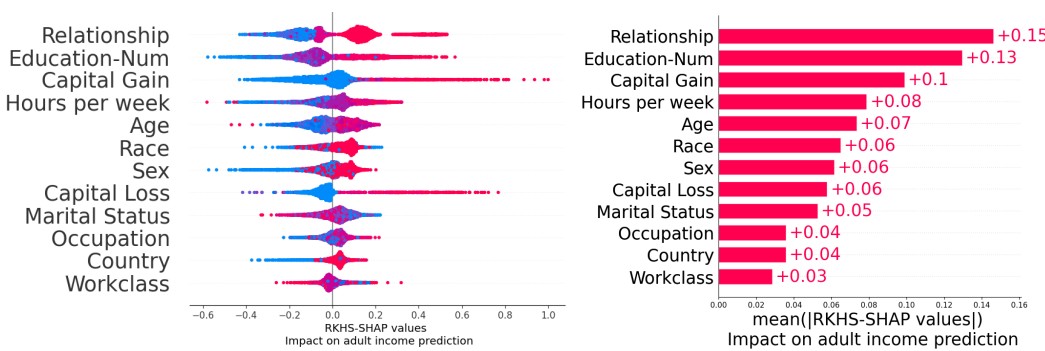

Figure 11: Beeswarm and barplot for the census income data prediction problem

that features such as relationship, education level and capital gain are most predictive of whether a person earns more $ 50k a year. We see that as a person grows older, it is more likely to earn more, but the effect is not as impactful as, e.g. Education level or Capital gain.

**League of Legends Win Prediction** Finally, we use the Kaggle dataset League of Legends Ranked Mathches which contains 1,800,000 players matches starting from 2014. We follow the preprocessing steps from [28], and obtained 71 features at the end. We deploy RKHS-SHAP to explain the fitted

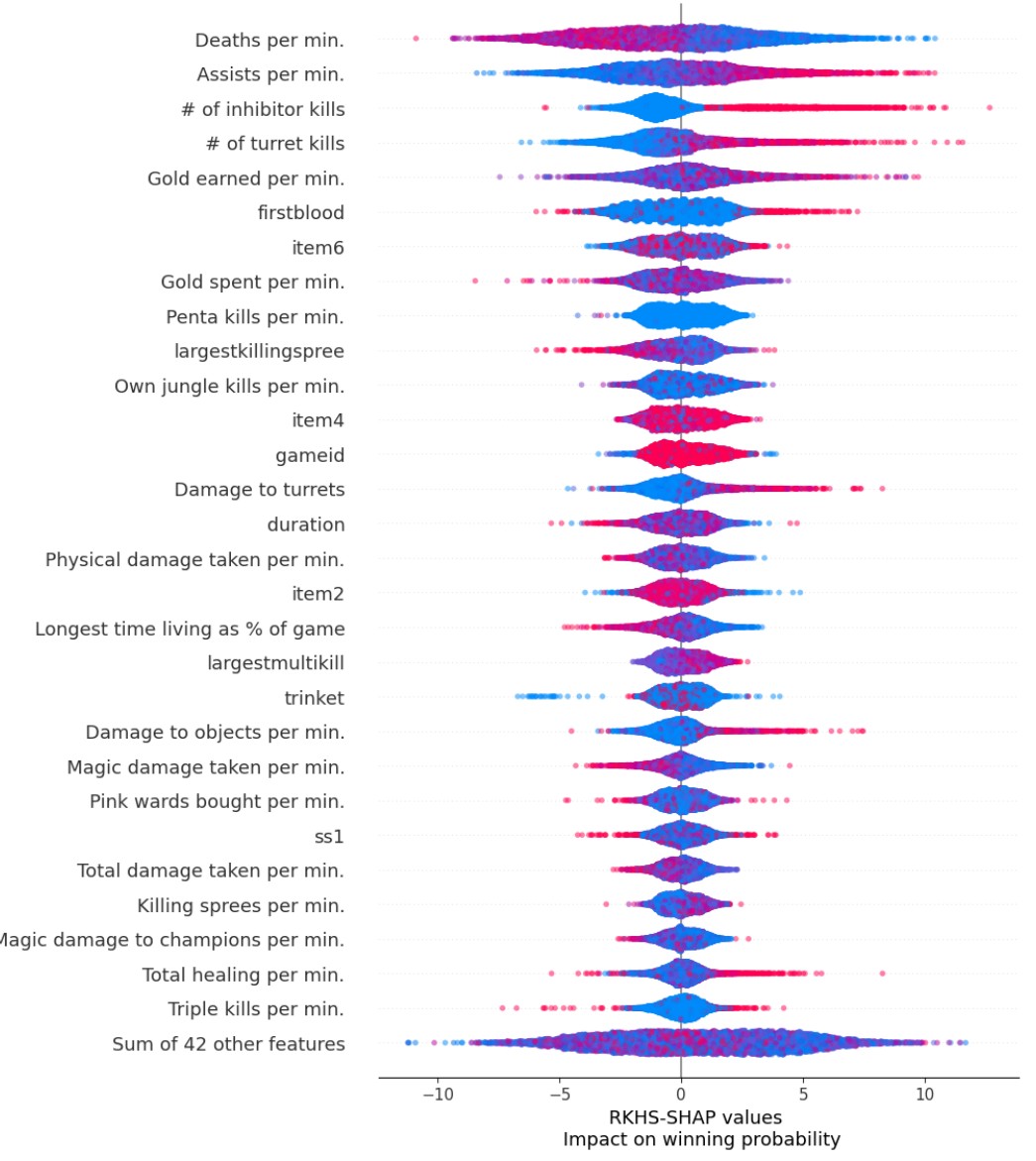

Figure 12: Beeswarm plot for the League of Legends player winning prediction problem obtained using RKHS-SHAP.

Kernel Logistic regression model and obtain results in Figure 12. We see that features such as "Deaths per min" and "Assists per min" are most influential to the match outcome. It follows the game mechanism, as a player is intuitively considered as "strong" if he doesn't die often in a round of the game. We would also like to point out we recover similar explanations from [28], where they applied TreeSHAP to recover the explanations, see Fig. 13. Interestingly, our kernel logistic regression seems to believe that "Gold earned per min" is less informative to the winning probability compared to "Deaths per min", which is different to the results obtained from the tree ensembles.

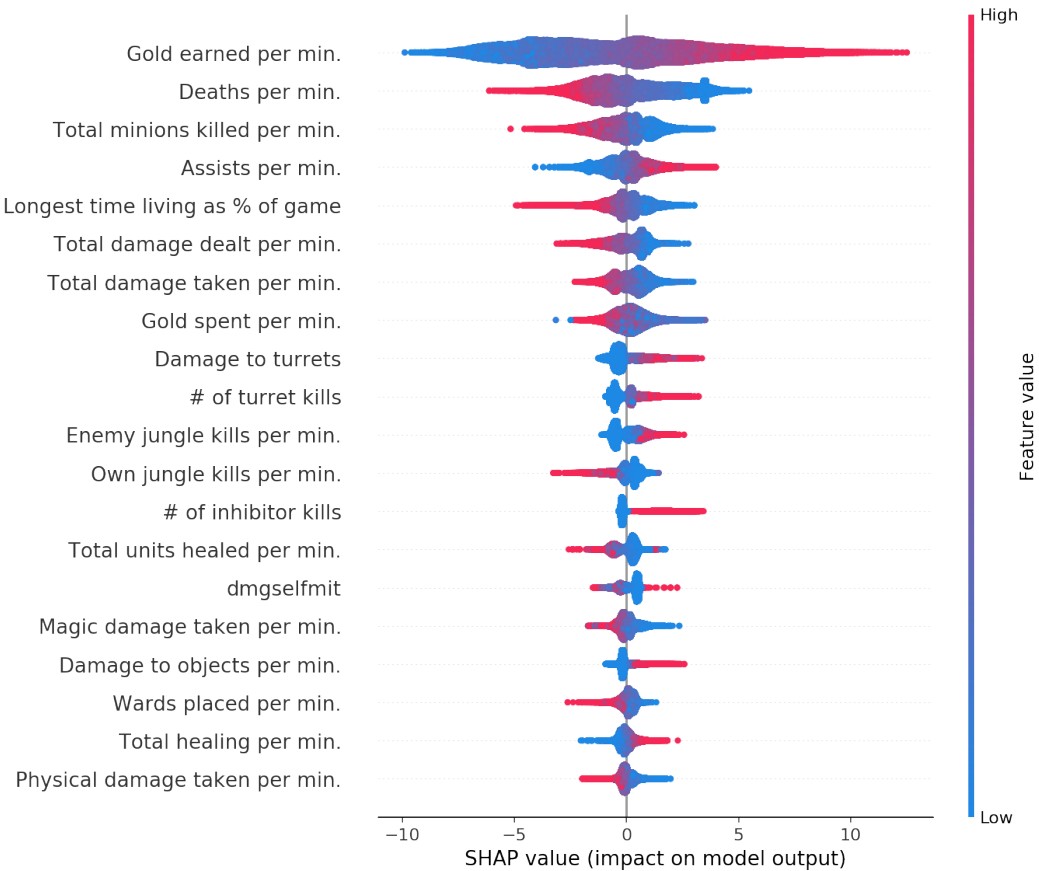

Figure 13: Beeswarm plot for the League of Legends player winning prediction problem obtained using TreeSHAP. Similar insights are recovered compared to RKHS-SHAP. However, since the two methods are explaining different models – an RKHS function and a tree, it is not possible to tell which one gives more "correct" explanation.