# OpenReview forum: "RKHS-SHAP: Shapley Values for Kernel Methods"
_NeurIPS.cc/2022/Conference — NeurIPS 2022 Accept_

### Official Review · Reviewer_iFp9 · 2022-07-10

**Rating:** 7
**Confidence:** 4
**Soundness:** 4 excellent
**Presentation:** 3 good
**Contribution:** 3 good

**Summary:**

This paper considers the calculation of Shapley value explanations in the context of kernel models. The idea is to leverage the structure of such models to generate better Shapley value estimates (e.g., more accurate or faster), similar to what TreeSHAP enables for tree ensembles.

The main technical insight has to do with how held-out features are handled. Rather than estimating $\mathbb{E}[f(X_{s^c}, x_s) | X_s = x_s]$ by sampling values from $p(X_{s^c} \mid X_S = x_S)$, or estimating $\mathbb{E}[f(X_{s^c}, x_s)]$ by sampling from $p(X_{s^c})$, the authors suggest using the underlying kernels to calculate the conditional mean embedding (CME) or kernel mean embedding (KME) for the held-out features (possibly conditioned on the observed ones). This may lead to more reliable and/or faster estimate of the value function (aka coalitional game), and the authors then suggest using KernelSHAP's weighted least squares algorithm for the final Shapley value approximation.

**Questions:**

If I understand correctly, the authors may have understated the computational cost of their approach, as well as the difficulty of achieving reliable CME estimates with realistic datasets. To elaborate:

1. Whether you're estimating interventional or observational Shapley values, each coalition/subset will require a separate evaluation of the value function (coalitional game). And for each evaluation, you'll need to estimate $\mu_{X_{s^c} \mid X_s = x_s}$ or $\mu_{X_{s^c}}$, which seems like a potentially expensive operation. (Well, the KME can be calculated once per feature, but not the CME.) Thus, RKHS-SHAP does not seem like it offers any run-time benefits, in fact it may make the estimation even slower. Am I missing something?
2. $\mu_{X_{s^c} \mid X_s = x_s}$ is not a simple thing to estimate, it would seem to require that you filter for examples in your dataset where $X_s = x_s$. In a high-dimensional dataset, or a dataset with continuous features, wouldn't the number of matching rows be very small, particularly for $s$ with large cardinality? Furthermore, if you somehow are able to find matching rows that help you estimate $\mu_{X_{s^c} \mid X_s = x_s}$, couldn't you use the same rows to directly estimate $\mathbb{E}[f(X_{s^c}, x_s) | X_s = x_s]$ even for non-kernel models? It seems like this trick only works in settings where, by assumption, you could reliably estimate the conditional expectation of the model output. (By the way, this approach for estimating the conditional expectation is discussed in Sundararajan et al., 2020.) Is this right, or am I missing something?

About the robustness results:
- Similar to my point under "weaknesses," can you elaborate on whether you believe this result is somehow specific to RKHS-SHAP, or just a property of Shapley values applied to kernel models?
- Could you comment on whether the constants in the robustness bounds are actually small? If not, these results may be meaningless. E.g., we could establish robustness bounds for Shapley values with arbitrary models whose predictions lie within a bounded range - are these results more useful than that?


**Limitations:**

There are no negative societal impacts to this work.

As for methodological limitations, I believe those are primarily in the speed and reliability of the CME estimation (as discussed above). If my concerns are accurate, then I don't think these points are currently adequately discussed.

**Strengths And Weaknesses:**

I should mention that I'm not an expert in kernel methods, so apologies in advance if I misunderstood any key parts of the method.

### Strengths:

- This paper is the first to my knowledge that develops a Shapley value approach specifically for kernel methods. The insights appear new and possibly useful in practice. The results are non-trivial and seem to require a strong understanding of kernel methods, which is not currently well represented in the explainable AI literature. (On the other hand, the key techniques may be inaccessible to many readers, but this is perhaps inevitable.)
- The method enables regularization of explanations during training, which is generally difficult to achieve with Shapley values due to the cost of their calculation.
- The background section is overall very nicely written.

### Weaknesses:

- I may be wrong on this, but I think the authors could say more about how expensive it is to estimate $\mu_{X_{s^c} \mid X_s = x_s}$ repeatedly throughout the KernelSHAP WLS approximation for different coalitions $s$. I've elaborated on this in the "questions" section below.
- Again I may be wrong on this, but I think the paper understates the difficulty of estimating $\mu_{X_{s^c} \mid X_s = x_s}$. I've elaborated on this in the "questions" section below.
- The robustness results do not seem to be a property of RKHS-SHAP, but of Shapley values themselves when applied to kernel models for which the authors' assumptions hold. It's an interesting result and non-trivial to show, but it perhaps shouldn't be claimed to be related to RKHS-SHAP.
- In discussing the conditional expectation estimate in Section 3, the authors write that they "utilise the arsenal of kernel methods to estimate the conditional expectations directly." I believe this is very similar (except for the focus on kernels) to the second method introduced by Frye et al. 2020 that has a very similar motivation - the supervised surrogate model. That method is easier to train than the conditional generative model, does not require Monte Carlo sampling, and has no risk of off-manifold examples because it bypasses the generative modeling task. Perhaps this deserves some discussion?
- I thought the paper's exposition could greatly benefit from disentangling the two main elements of estimating Shapley values (and where RHKS-SHAP is helpful): 1) handling predictions with held-out features, and 2) accurately approximating Shapley values. Section 2 made it sound like this work would present a significant improvement over KernelSHAP (see lines 114-130), which led me to believe the improvement had to do with 2). Ultimately, it had to do with 1), which I don't understand as really being part of KernelSHAP. The main idea behind KernelSHAP is approximating Shapley values via weighted least squares, and the authors actually re-use this part. What's modified here is the sampling trick for handling held-out features, which isn't unique to KernelSHAP and also used by other Shapley value methods (e.g., IME).

A couple small things:
- On line 125, the formula for the closed-form solution to the WLS problem solved by KernelSHAP seems a bit informal due to the presence of $\infty$ entries in $W$
- It may be helpful to say explicitly that the interventional and observational coalitional games can be estimated via the KME/CME specifically because of the model's linear dependence on the features $\psi_x$. Of course, what we really want is the expectation of the model's output, and it's a unique property of such methods that we can get that via the expectation of the $\psi_x$ component.

There are a couple related works that seem like they should/could have been discussed here:
- The various ways of estimating $\mathbb{E}[f(X) \mid X_s = x_s]$ are discussed in a recent review paper [1], including parametric assumptions, generative modeling, and several more. The conditional generative model is not, I think, widely understood as a viable approach
- The idea of "attribution priors" [2] has been discussed as a way to regularize models' dependencies during training, although it's mainly used with gradient-based methods rather than Shapley values. Regularizing explanations is a natural idea, but Shapley values are somewhat unique in making this intractable
- ShapNets [3] also provide a fast Shapley value calculation, albeit for a specific modified DNN architecture, and one of the motivations was to enable explanation regularization. RKHS-SHAP provides something similar, but for kernel models

[1] Covert et al., "Explaining by removing: a unified approach to model explanation" (2021)

[2] Erion et al., "Improving permeance of deep learning models with axiomatic attribution priors and expected gradients" (2021)

[3] Wang et al., "Shapley explanation networks" (2021)

---

> ### Author Response · Authors · 2022-08-01
> **Responses to IFp9's reviews**
>
> ### General comments:
> We want to thank the reviewer for their detailed constructive comments and suggested references, which are valuable and appreciated. We also really appreciate the reviewer's suggestion on both narrative and literature to include, especially the ones on attribution prior, we will include them in the camera ready version.
>
> We would like to begin by addressing what appears to be a misunderstanding of the framework of Conditional Mean Embeddings. It is possible that several reviewer's concerns stem from this misunderstanding and are thus not valid.
>
> We hope to clarify the comments individually and hope the reviewer will increase the score afterwards.
>
> ### Specific comments:
>
> ---
> *"Whether you're estimating interventional or observational Shapley values, each coalition/subset will require a separate evaluation of the value function (coalitional game). And for each evaluation, you'll need to estimate  KME or CME, which seems like a potentially expensive operation..."*
>
> A: Please refer to our general comment on CME estimation and complexity analysis. In short, Conditional Mean Embedding for each coalition can be estimated with an $\mathcal{O}(n \sqrt{n})$ operation, and it is certainly an improvement over other methods, for example, over conditional density estimation. We have provided detailed complexity comparisons in Appendix A.
>
> ---
> *"CME is not a simple thing to estimate, it would seem to require that you filter for examples in your dataset where $X_S= x_S$. In a high-dimensional dataset, or a dataset with continuous features, wouldn't the number of matching rows be very small, particularly for s with large cardinality?...."*
>
> A: This is not right. As detailed in the general comments, CME is not estimated by filtering out dataset as the reviewer has suggested. In fact, CME is estimated via a vector-valued regression problem, and thus we do not require multiple values observed over the same conditioning value. Instead, we can utilise the smoothness across the conditioning variable. This approach has been widely adopted and shown to be effective in a wide range of applications, as detailed in the general comments.
>
>
> ---
> *"In discussing the conditional expectation estimate in Section 3, the authors write that they "utilise the arsenal of kernel methods to estimate the conditional expectations directly." I believe this is very similar (except for the focus on kernels) to the second method introduced by Frye et al. 2020 that has a very similar motivation - the supervised surrogate model...""*
>
> A: Thank you for this suggestion. In fact, we have mentioned Frye's approach in line 210-214. It is, however, worth noting that a recent paper by Yeh et al. 2022 raises some significant concerns regarding the approach by Frye et al., regarding the validity of their approach, e.g. the optimal regression based learner will yield constant explanations when explaining feature values that are unique in the dataset.
>
> ---
>
> *"Could you comment on whether the constants in the robustness bounds are actually small? If not, these results may be meaningless. E.g., we could establish robustness bounds for Shapley values with arbitrary models whose predictions lie within a bounded range - are these results more useful than that?"*
>
> A: The constant in bound for observational SV represents the (upper bound of the) smoothness of the CME functions $\mu_{X_{S^c} \mid X_S}$ for all S. Therefore, when the features have stronger non-linear dependencies, the norm of their CME function will be larger, thus the bound will be less tight. We believe this to be quite natural because features having stronger non-linear dependencies means their conditional density is more difficult to capture, thus harder to bound them: the more difficult the problem gets, the harder it is to get a tight bound. We can include a further discussion on this in the camera-ready version.
>
> We also wanted to highlight, as mentioned in line 284-285, this assumption has also been adopted in Park et al. 2020 for their analysis.
>
> [Park et al. 2020] A measure-theoretic approach to kernel conditional mean embeddings.
>
> ---
>
> *"Similar to my point under "weaknesses," can you elaborate on whether you believe this result is somehow specific to RKHS-SHAP, or just a property of Shapley values applied to kernel models?"*
>
> A: We believe this result is specific to RKHS-SHAP. RKHS-SHAP method provides an estimator of the observational/marginal value function using CME/KME tools, which we then use to provide the the robustness analysis. When Shapley values are applied to kernel functions more generally, prior to our work, the only way to estimate them in the observational setting was to perform conditional density estimation, which would require different tools for robustness analysis, depending on the particular conditional density estimation method used.
>
> ---
>
> *We hope the reviewer could increase their score as the concerns are now clarified. Thank you very much!*

---

> > ### Comment · Reviewer_iFp9 · 2022-08-07
> > **About CME estimation and relationship to Frye et al.**
> >
> > Thanks to the authors for their response. I believe I understand the method better now, particularly the approach of casting CME estimation as a regression problem. I have some other comments, but perhaps we can first resolve a couple remaining questions about CME estimation.
> >
> > In lines 176-190 of the original work, I was unaware that the authors would use this approach for feature subsets $X_S$ because no feature subset notation is used. Can this be amended in the revision? And what modifications are necessary in the equation for $\hat \mu_{Y \mid X}(x)$ when we instead want $\hat \mu_{Y \mid X_S}(x_S)$?
> >
> > It seems like this approach actually ties in quite closely with that of Frye et al., more so than the paper currently reflects or I previously realized. In both cases, there's a random variable $Z$ jointly distributed with $X$, and both papers are interested in the conditional expectation $\mathbb{E}[Z | X_S = x_S]$; both papers approximate it by minimizing a least-squares-like loss function, or $\min_g \mathbb{E}[ ||Z - g(X_S) ||^2]$. The differences seem to be that 1) this work adopts $Z = \psi_Y$ and uses $||\cdot||_{\mathcal{H}_g}^2$ while Frye et al. adopts $Z = f(X)$ and uses MSE, and 2) this work lets $g$ be a kernel method while Frye et al. use a neural network. Can the authors comment on this, are these approaches as similar as they seem to me?
> >
> > Not that I think the similarity poses a novelty issue, I believe there's room for a method designed specifically for kernel methods. However, acknowledging the similarity would be helpful, and I think the authors must reconsider whether their current critiques of Frye et al.'s supervised method also hold for their own approach. Specifically, if Frye et al.'s method risks reverting to the empirical conditional expectation (to use the term from Sundararajan et al.), does this method not risk the same issue? If not, is it saved by the kernel method's relative lack of flexibility compared to a neural network? Or does the regularization term (line 182) play a crucial role that is not currently explained? And do the authors believe that similar regularization tactics are somehow not applicable to Frye et al.'s deep learning-based approach?

---

> > > ### Author Response · Authors · 2022-08-08
> > > **Responses to iFp9 on CME and comparison with Frye et al.**
> > >
> > > Thank you for your reply. We summarise our response as follows:
> > >
> > > ### CME Part
> > > Thanks for the suggestion for the amendments which will improve clarity. Since the original discussion was introducing the general concept of CME, we did not include subset notation, to keep the notation simple.
> > >
> > > We delayed the introduction of CMEs with feature subsets until Proposition 2 and provided empirical estimation methods in Proposition 3. In the camera-ready version with fewer space constraints, we shall add an extra sentence in line 191 to briefly introduce the concept and expand later in section 3.
> > >
> > > To obtain $\mu_{Y\mid X_S}$, one simply computes the kernel matrix for features of $X$ in $S$, instead of computing overall features. Please see proposition 2 for more details.
> > >
> > > ---
> > >
> > > ### Comparison with Frye et al.
> > >
> > > We agree with the reviewer that the similarities and differences between Frye's approach and ours are worth emphasizing in the paper because they share a similar regression-like intuition, but, as we highlight below, differ in most other aspects:
> > >
> > > - **Difference in regression target:** Frye's approach regresses onto scalar values $f(X)$ for a specific $f$ while our approach regresses onto an infinite dimensional feature map $\psi_{X_{S^c}}$. Note that our model is aiming to capture representation of the full conditional distribution (via its RKHS embedding) rather than the conditional expectation for a specific $f$.
> > >
> > > - **Difference in dependency on $f$:** CME estimation depends on the function space that $f$ belongs to and not on the specific $f$. This subtle but crucial point allows one to apply Shapley functionals as attribution priors during the learning of $f$ itself, in order to regularize it.
> > >
> > > - **Difference in hypothesis space:** Frye's approach uses a scalar-valued parametric neural network model, while our approach uses an RKHS-valued non-parametric kernel ridge regression with a ridge penalty to promote smoothness.
> > >
> > > - **Difference in learning procedure:** For each $S$, Frye's approach needs to relearn the parameters of a new neural network. On the other hand, assuming that the function space where $f$ lies is fixed (which is certainly true after learning $f$ itself), our method has a closed form ridge regression solution with no additional parameters to be learned (i.e. all kernel lengthscales for CME estimations are already determined by the function space of $f$).
> > >
> > > Because of all the differences outlined above, we believe that our approach does not suffer from the risk of reverting back to an empirical conditional expectation $\nu^{CE}_{x, f} = \frac{\sum_{i=1}^n\textbf{1}[{x_i}_S = x_S]f(x_i)}{\sum_{i=1}^n\textbf{1}[{x_i}_S=x_S]}$ compared to Frye's deep method, because it promotes smoothness in the conditioning variable due to its regularisation. Therefore, even when we are explaining new feature values that have not been seen before, we would not get constant explanations. This is illustrated by our experiments in Appendix D2 where we do not see clusters of Shapley values even though we are explaining on withheld data points.
> > >
> > > We hope the reviewer could increase their score if their concerns are now clarified. Thank you very much!

---

> > > > ### Comment · Reviewer_iFp9 · 2022-08-09
> > > > **Thanks for clarification; other questions**
> > > >
> > > > Hi, thanks for the clarification about Propositions 2 and 3. This part of the paper was a bit hard for me to follow, and I suspect that providing a bit more explanation would be helpful to other readers as well.
> > > >
> > > > **About the Frye et al. similarities.** Like you said, I think these comparisons would be helpful to discuss in the paper. There are a couple points I still find myself disagreeing with though:
> > > >
> > > > - Frye et al.'s method doesn't require training a separate neural network for every feature subset $S$. That would be very slow and impractical, so a single model is instead trained with randomly sampled masks (see eq. 17 [here](https://openreview.net/pdf?id=OPyWRrcjVQw)). This is more computationally efficient, and it also functions as a form of regularization by making it more difficult for the model to memorize labels.
> > > > - If I understand correctly, the main argument for your CME approach not being susceptible to reproducing the empirical conditional expectation is that the ridge penalty encourages smoothness. Why can't we use comparable regularization tactics in the Frye et al. deep learning approach? I suppose main point here is that *neither method is actually susceptible to this risk*. The flaw presented in Yeh et al. (2022) seems exaggerated, it's based on a worst-case memorization scenario that should be straightforward to mitigate with standard regularization tactics and early stopping based on a validation loss. Note that a similar argument could be made with any use of empirical risk minimization (i.e., that the model will collapse to the empirical expectation), but DNNs trained via ERM tend to generalize just fine on out-of-sample data. If it's helpful, I can even point to papers that successfully used Frye et al.'s supervised surrogate approach without obtaining constant explanations.
> > > >
> > > > **About whether the robustness result is specific to RKHS-SHAP.** This topic was addressed in your response, but I'm not sure we're on the same page yet. The fact that you used CME/KME as tools to analyze the robustness does not mean that conditional Shapley values' robustness is due to the RKHS-SHAP estimation approach. The estimation strategy does not affect the properties of the quantity being estimated, and these are equivalent to conditional Shapley values, right? The robustness result still seems like an intrinsic property of Shapley values when applied to this type of model; in other words, the property holds regardless of how the Shapley values are calculated (e.g., using your approximation, or using oracle access to the conditional distributions). Can you please clarify if I'm missing something?
> > > >
> > > > **Writing.** There was one more weakness from my review that I would ideally like to see incorporated in the revisions - the one beginning with "I thought the paper's exposition could greatly benefit from disentangling the two main elements of estimating Shapley values." This discussion has reinforced my understanding that the main ideas in this work are related to how held-out features are handled. This was surprisingly hard to grasp from reading the paper due to repeated comparisons with KernelSHAP (which is focused primarily on how to handle the exponential complexity), so it would be helpful to rework early parts of the paper explaining what the contributions are and how it relates to prior work. For context, see this recent review paper that nicely distinguishes these two aspects of Shapley value estimation [1].
> > > >
> > > > [1] Chen et al., "Algorithms to estimate Shapley value feature attributions" (2022)
> > > >
> > > > This is my last set of issues to discuss, I expect to be able to finalize my score after we resolve these. Sorry we're getting close to the deadline for author-reviewer discussion.

---

> > > > > ### Author Response · Authors · 2022-08-09
> > > > > **Response**
> > > > >
> > > > > **Frye et al similarities:** We thank the reviewer for clarification on Frye's approach. We agree with the reviewer that by promoting smoothness during learning, Frye's approach should not arrive to the overfitting situation Yeh et al. (2022) mentioned. We will make subsequent changes to the camera ready version on Frye's approach and emphasize more on the conceptual similarities between ours and theirs. Thank you again for pointing that out explicitly.
> > > > >
> > > > > ---
> > > > >
> > > > > **Robustness results:** We believe this confusion is caused by us having a different view on RKHS-SHAP's value function estimator. Our approach certainly utilised properties of RKHS functions to model the value functions using KME/CMEs, where we first propose a population level model in Prop 2, and a finite-data estimator in Prop 3. However, had one choose to model value functions differently, for example, proposing another conditional mean (population level) estimator, the subsequent analysis will be very different. This can be seen e.g. by the bound in theorem 6, for Observational Shapley functionals, the upper bound of the norm of the population CME function appears.
> > > > >
> > > > > Because of this, we still believe the robust results is still RKHS-SHAP specific because we made a specific modelling choice of the conditional expectation and the analysis is then built on that choice we made.
> > > > >
> > > > > ---
> > > > >
> > > > > **Writing:** We thank the reviewer for their suggestion. We agree that more emphasize can be put on highlighting the two main elements of estimating Shapley values. We promise to make changes in the early parts of the paper to strengthen this point to improve reading experience.
> > > > >
> > > > > ---
> > > > >
> > > > > We really appreciate the reviewer's effort to improve the clarity of the paper. We promise to make the following changes in the camera ready version:
> > > > > 1. Restructure the narrative in early parts of the paper to emphasise the aspect we contributed to Shapley value estimation for RKHS functions
> > > > > 2. We will change our comments on Frye's approach and draw parallels to the conceptual similarities among the two.
> > > > > 3. Add a few lines when introducing CME explaining the intuition behind.
> > > > >
> > > > > We hope we have resolved most of the reviewer's question and hope they could increase their scores. Thank you very much.

---

> > > > > > ### Comment · Reviewer_iFp9 · 2022-08-09
> > > > > > **Thanks**
> > > > > >
> > > > > > Sounds good, discussing these points in the paper will resolve all my remaining concerns. I'm going to raise my score.

---

> > > > > > > ### Author Response · Authors · 2022-08-09
> > > > > > > **Thanks**
> > > > > > >
> > > > > > > Thank you very much!

---

### Official Review · Reviewer_YvsH · 2022-07-13

**Rating:** 4
**Confidence:** 3
**Soundness:** 3 good
**Presentation:** 3 good
**Contribution:** 2 fair

**Summary:**

- The authors proposed a feature attribution method for explaining kernel methods.
- Some analysis has been provided to demonstrate the robustness of the proposed methods.
- Experiments are carried out to show the effectiveness of the method.

**Questions:**

- Since kernel methods are differentiable, one can use gradient based methods for understanding kernel methods. How is that compared to Shapley based methods?

**Ethics Review Area:**

["I don’t know"]

**Limitations:**

- Computational complexity.
- Only applicable to kernel methods.

**Strengths And Weaknesses:**

Strengths:
- The robustness of the method under local perturbation has been rigorously verified, which is quite important in such explanation methods.
Weaknesses:
- The proposed method falls short of computational complexity. It seems not practical.
- The established theorem is relatively straightforward. It is obvious from the definition.

---

> ### Author Response · Authors · 2022-08-01
> **Responses to YvsH's reviews**
>
> *"Weakness: Only applicable to Kernel methods"*
>
> A: The developed method follows the well established line of research on explainability tools for specific function models: TreeSHAP for tree based models, DeepSHAP for deep models, LinearSHAP for linear models etc. By focusing on a specific function model, more efficient and performant approximation algorithms can be developed, as showcased in our paper -- which focuses on kernel methods, for which model-specific tools for estimation of Shapley value have not previously been considered.
>
> Therefore, our contribution fills an important gap in the literature and we do not believe the focus on kernel methods to be its weakness.
>
> ---
> *"Weakness: The proposed method falls short of computational complexity. It seems no practical."*
>
> A: Please refer to the general responses where we addressed and highlighted the computational complexity of our algorithm. In short, our algorithm takes $\mathcal{O}(n\sqrt{n})$ time when computing the conditional mean embeddings.
>
> To showcase the large scale capabilities of our algorithm, we have ran our experiments on a "league of legends win prediction" and reported the results in Appendix D2, where we explained a kernel logistic regression with $1,800,000$ instances with $71$ features under $10$ minutes, demonstrating the practicality and scalability of our algorithm.
>
> ---
> *"The established theorem is relatively straightforward. It is obvious from the definition."*
>
> A: Could you please clarify what do you mean by straightforward? and which part follows directly from which definition?
>
> ---
> *"Since kernel methods are differentiable, one can use gradient based methods for understanding kernel methods. How is that compared to Shapley based methods?"*
>
> A: A: While gradient-based and Shapley value-based attributions are both trying to provide explanation to an algorithm, the ways they define the notion of "importance" are very different. In particular, the popular Integrated Gradient (IG) method [Sundararajan et al. 2017] was shown to theoretically approach Aummann-Shapley value as discussed in [Chen et al. 2019], a fundamentally different concept to Shapley values.
>
> This difference leads to IG failing to satisfy some desirable feature attribution axioms that Shapley values do satisfy. For example, when feature i and j contributed equally to the function f at all coalition S, i.e. $\nu_f(\{i\}\cup S) = \nu_f(\{j\}\cup S)$ with $\nu_f$ the value function defined with respect with $f$, IG do not necessarily returns the same attribution score to features i and j, but SVs would. Moreover, when feature i does not contribute to the function f at all, the attribution score from IG will not always be 0 while Shapley value based approach would. See examples from this article [1] for further reference.
>
> - [1] https://towardsdatascience.com/limitations-of-integrated-gradients-for-feature-attribution-ca2a50e7d269
> - [Sundararajan et al. 2017] Axiomatic attributino for deep networks
> - [Chen et al. 2019] Explaining Models by Propgagating Shapley values
>
> ---
>
> We hope the reviewer could increase their score as these concerns are now clarified and in light of other reviewer's positive comments. Thank you very much.

---

> > ### Comment · Reviewer_YvsH · 2022-08-07
> > **Comparison with IG**
> >
> > Hi, thanks for response. I was looking for a comparison with IG empirically.

---

> > > ### Author Response · Authors · 2022-08-08
> > > **Response to comparison with IG**
> > >
> > > Empirical comparison between RKHS-SHAP and kernel method specific Integrated Gradients is outside of the scope of our contribution and, for the reasons we outline below, would not be particularly meaningful even if it was possible to conduct in this limited time scope. However, as RKHS-SHAP is a kernel method specific estimation way to compute Shapley values, all existing discussion and comparisons of Shapley values vs Integrated Gradients will still be applicable to our discussion.
> > >
> > > The original author of Integrated Gradients (IG), Mukund Sundararajan, in his work "The Many Shapley Values for Model Explanation" Remark 4.5, constructed an example where they compared IG with a variant of Shapley values called BSHAP, and showed that both attributions are "intuitively reasonable" but "not immediately clear which interpretation is obviously superior".
> > >
> > > There is no consensus in the literature on the preference between Shapley values and Integrated Gradients. Both methods satisfy different sets of axioms and offer attribution with different interpretations. As an example, in article [1], the author gave an example where IG fails to satisfy several Shapley's desirable axioms. In particular, IG can assign different attributions to two features that always have the exact same effect on the model, and can assign positive attributions to features that have no effect on the model.
> > >
> > > In summary, the discussion on Integrated Gradients vs Shapley values, as two different explainability approaches, is interesting, but unrelated to our contribution. We focused on providing a kernel method specific Shapley value estimation scheme, following the established line of research on designing model-specific SHAP algorithms such as linearSHAP, treeSHAP, and DeepSHAP. While one could conduct a study on IG vs Shapley values, but that will be a very different paper to ours.
> > >
> > > [1] https://towardsdatascience.com/limitations-of-integrated-gradients-for-feature-attribution-ca2a50e7d269

---

> > > > ### Comment · Reviewer_YvsH · 2022-08-08
> > > > **Empirical evaluation D2**
> > > >
> > > > Hi, I more than agree with you about the axiomatic difference between the two algorithms. I am talking about the form of comparison in D.2. Here you have already coded the experiments to find important features with the RKHS-SHAP, so the comparison of such should be straightforward.
> > > >
> > > > Concretely, suppose one has two algorithms RKHS-SHAP and IG, and ranks features by their importance provided by each of the algorithm. Can you show that by masking these features with their average in the order, which algorithm leads to a faster decreasing in classification accuracy / MSE?

---

> > > > > ### Author Response · Authors · 2022-08-09
> > > > > **Response to comparing IG with Shapley values**
> > > > >
> > > > > Thank you for your suggestion. We have included the extra experiment you requested in the supplementary materials, the PDF file is called “RKHS_SHAP_versus_IG_comparison_for_R2”. Please have a look.
> > > > >
> > > > > We hope this will close the discussion on the topic Shapley values versus Integrated Gradients because we still believe it to be outside the scope of our contributions.
> > > > >
> > > > > We hope the reviewer can now increase their scores as we have fully answered and clarified all your queries and concerns.
> > > > >
> > > > > Thank you.

---

> > > > > > ### Comment · Reviewer_YvsH · 2022-08-09
> > > > > > **local vs global**
> > > > > >
> > > > > > Hi thanks for the efforts. The experiments do not 100% match what I described above. "As Riberio et al 2016 showed,
> > > > > > global and local importance do not necessarily imply each other." Yes I agree with the above and that is where the mismatch comes from.
> > > > > >
> > > > > > I am NOT asking for RETRAINING the model every time a feature is removed.
> > > > > >
> > > > > > The experiment is simpler than it:
> > > > > >
> > > > > > Inputs: We are presented a trained model to explain.
> > > > > >
> > > > > > - For a given example, we apply a method (IG/RKHS Shapley), we get importance scores.
> > > > > > - We rank features by importance scores.
> > > > > > - We feed into the SAME model with masked samples. Masks are created by the rank of importance scores. The mask means replacing that feature value with its mean, or some null value.
> > > > > > - On the test data, we get the accuracy / loss, as the function of the number of masked features.
> > > > > >
> > > > > > Outputs: We plot the figure of this function.
> > > > > >
> > > > > > The method serves as a heuristic evaluation of the LOCAL importance. The con here is that replacing is never the best we can do, and the model may not understand that (which means a bias is introduced). A more accurate method is to get the conditional distribution of that feature conditioned on the value of other features of each sample, theoretically. It can be approximated by the unconditional distribution of the feature in-sample if we assume independence between features. Of course it seems to be too complicated to be included here, and often times a simple masking by the mean can give a good estimate.

---

> > > > > > > ### Author Response · Authors · 2022-08-09
> > > > > > > **Thank you for the suggestion**
> > > > > > >
> > > > > > > Thank you for the suggested further experiments, however we don't have the capacity to run them given the limited time left.
> > > > > > >
> > > > > > > We sincerely hope the reviewer can consider increasing their score as we have now:
> > > > > > >
> > > > > > > - Clarified the practicality of our method (can scale up to n=1,800,000 and applied to 71 features)
> > > > > > > - Clarified the point on IG vs Shapley values with experiments and theoretical discussions.
> > > > > > > - Explained why designing kernel method specific SHAP approximation is a well-motivated piece of research.
> > > > > > >
> > > > > > > Thank you very much.

---

> > > > > > > > ### Comment · Reviewer_YvsH · 2022-08-10
> > > > > > > > **When is deadline at authors side**
> > > > > > > >
> > > > > > > > Hi when is the deadline at your side? At my side, it seems I am able to make changes until Aug 19th. Is it that you will not be allowed to make changes after today? How about updating supplement materials? It is also fine to include some annoymous github link which is to be updated before Aug 19th if you prefer.

---

> > > > > > > > > ### Author Response · Authors · 2022-08-10
> > > > > > > > > **Reply query on author deadline**
> > > > > > > > >
> > > > > > > > > We will forward the message we recieved from the Program Chairs:
> > > > > > > > >
> > > > > > > > >
> > > > > > > > > - During this decision making phase, the authors are not allowed to interact with the reviewers, area chairs nor senior area chairs. Reviewers, area chairs and senior area chairs will work with what they have at this moment, including your manuscript, supplementary material, reviewers' reviews, your rebuttals and any discussion between reviewers and you, in order to collectively arrive at the recommendation of your submission.
> > > > > > > > > With the unprecedented number of submissions as well as unprecedented number of reviewers we have this year, it has not been without hiccups, but we anticipate that we would be able to release decision notification on September 14, as was planned originally.
> > > > > > > > > If your submission is accepted, you will have another month to prepare the camera-ready version.
> > > > > > > > > Sincerely,
> > > > > > > > > Program Chairs
> > > > > > > > >
> > > > > > > > > ---
> > > > > > > > >
> > > > > > > > > We will consider extra experiments that reviewer suggested during the preparation of the camera ready version.

---

### Official Review · Reviewer_xJQh · 2022-07-16

**Rating:** 6
**Confidence:** 3
**Soundness:** 3 good
**Presentation:** 3 good
**Contribution:** 3 good

**Summary:**

This paper proposed a novel method for calculating Shapley value for kernel-based models. The conditional expectation-based kernel method suffers from the error caused by the sampling-based conditional expectation estimation, while this paper utilize the property of kernel method provided non-parametric estimater of the value function of Shapley value directly. They theorectically showed the robustness of RKHS-SHAP. They also propose Shapley regulariser which could control specific features' contributions to the model. Extensive experiments verified their methods.

**Questions:**

N/A

**Ethics Review Area:**

["I don’t know"]

**Strengths And Weaknesses:**

[Strength]
1. This paper rigorously proved the non-parametric estimater of the value function for kernel-based method.
2. Extensive experiments verified the effectiveness and robustness of the proposed methods.

[Weakness]
1. The method provided in this paper is only applicable to the kernel-based method.
2. The paper only considered the conditional/marginal expectation-based Shapley value, while lacked of discussion for baseline value-based Shapley value (BShap) [cite 1].

[cite 1] Mukund Sundararajan and Amir Najmi. The many shapley values for model explanation. In International Conference on Machine Learning, pages 9269–9278. PMLR, 2020.

---

> ### Author Response · Authors · 2022-08-01
> **Responses to xJQh's review**
>
> *"Weakness: The method provided in this paper is only applicable to the kernel-based method"*
>
> A: The developed method follows the well established line of research on explainability tools for specific function models: TreeSHAP for tree based models, DeepSHAP for deep models, LinearSHAP for linear models etc. By focusing on a specific function model, more efficient and performant approximation algorithms can be developed, as showcased in our paper -- which focuses on kernel methods, for which model-specific tools for estimation of Shapley value have not previously been considered.
>
> Therefore, our contribution fills an important gap in the literature and we do not believe the focus on kernel methods to be its weakness.
>
> ---
>
> *"Weakness: The paper only considered the conditional/marginal expectation-based Shapley value, while lacked of discussion for baseline value-based SV (BSHAP)"*
>
> A: In lines 90-91, we discussed a general formulation of value functions for ML explanation, which composes of an expectation of f with respect to some reference distribution. In line 98-99, we gave two other examples that are not marginal/conditional, to showcase the literature, and explained in line 100 why we chose to focus on conditional/marginal SVs -- precisely because they are the two most commonly discussed variants, following the work from Lunberg et al. 2017, Janzing et al. 2019, Chen et al. 2020, Frye et al. 2021, Yeh et al. 2022.
>
> We also did not discuss BSHAP for the following practical reason: it imputes pre-defined values to represent missingness in a value function. However, besides in obvious cases such as in imaging, where imputing 0 corresponds to blacking out a pixel, it is not obvious how one should pick a baseline to impute in practice. Imputing the mean value simply corresponds back to the marginal expectation in our case.
>
> Ref:
> - [Lundberg et. al. 2017] A Unified Approach to Interpreting Model Predictions
> - [Janzing et. al. 2019]: Feature relevance quantification in explainable AI: A causal problem
> - [Chen et al. 2020]: True to the Model or True to the Data
> - [Frye et al. 2021] Shapley explainability on the data manifold
> - [Yeh et al. 2022] Threading the Needle of On and Off-Manifold Value Functions for Shapley Explanations
>
> ---
>
> We hope the reviewer could increase their score as these concerns are now clarified. Thank you very much!

---

### Author Response · Authors · 2022-08-01
**General Comments**

We thank the reviewers for their comments. We are encouraged to see that all reviewers found our work interesting and important. We first give a general discussion regarding points raised by multiple reviewers, mainly on the CME estimation [R3] and computational complexity of RKHS-SHAP [R2-R3], and then address each reviewer's comment individually.

---

### Conditional Mean Embeddings [R3]

Estimating the conditional mean embedding (CME) of $p(Y\mid X=x)$, defined as $\mu_{Y\mid X=x} = \mathbb{E}[\ell(Y, \cdot)\mid X=x] \in H_\ell $ for some kernel $\ell$ on $\mathcal{Y}$, does not require a sample of multiple observations from the conditional distribution $p(Y\mid X=x)$. Hence, the CME is not estimated by filtering out dataset as R3 suggested. Instead, we only require a sample from the joint distribution of $(X,Y)$ and use a vector-valued non-parametric regression approach (see lines 176-190). This allows us to utilise smoothness across the conditioning variable. In detail, the general finite-data CME $\mu_{Y\mid X=x}$ estimator takes the following form:

$$\hat{\mu}_{Y\mid X}(x) =  \Psi_\mathbf{y}(\mathbf{K}_\mathbf{XX} + n\eta \mathbf{I})^{-1}\Psi_\mathbf{X}^\top \psi(x)$$

as shown in line 190 in the main text. Here, $\Psi$ denotes the corresponding feature matrices, which can be different for $X$ and $Y$, and $\mathbf{K}$ the kernel matrices. The theoretical justification of this formulation has been studied rigorously in this decade by papers such as [Grünewälder et al. 2012], [Klebanov et al. 2019], and [Park et al 2020]. There are also key results showing that this mapping of distribution to RKHS is injective when using characteristic kernels (e.g. RBF) as well. [Sriperumbudur et al. 2011]

From the practical perspective, CMEs have been widely adopted for various ML tasks that require a rigorous representation of conditional densities, such as the work in [Ton et al. 2021] where they have used CME for conditional density estimation under meta learning setting, or the work in [Chau et al 2021] where CME is used for two-staged regression to handle multiple modality. [Park et al. 2021] also used CME to test for conditional distributional treatment effect. The lists of CME application goes on and [Muandet et al 2017] is an excellent reference for this topic.

References:
- [Sriperumbudur et al. 2011] Universality, Characteristic Kernels and RKHS Embedding of Measures
- [Grünewälder et al. 2012]: Conditional Mean Embeddings as Regressors
- [Klebanov et al. 2019]: A Rigorous Theory of Conditional Mean Embeddings
- [Park et al 2020]: A Measure-Theoretic Approach to Kernel Conditional Mean Embeddings
- [Ton et al 2021]: Noise Contrastive Meta-Learning for Conditional Density Estimation using Kernel Mean Embeddings
- [Chau et al 2021]: BayesIMP: Uncertainty Quantification for Causal Data Fusion
- [Park et al. 2021]: Conditional Distributional Treatment Effect with Kernel Conditional Mean Embeddings and U-statistic Regression
- [Muandet et al. 2017]: Kernel Mean Embedding of Distributions: A Review and Beyond

-----

### Complexity conerns [R2-R3]

While naive estimation of CME requires $\mathcal{O}(n^3)$ due to inversion, there are numerous large-scale kernel approximation technique one could choose from to reduce the complexity as discussed in lines 187-190. In this work, we use FALKON [Giacomo et al. 2020], a Nyström based preconditioner for conjugate gradient descent to compute the heaviest Kernel matrix vector multiplications in the paper. This allows us to reduce the computational complexity from $\mathcal{O}(n^3)$ to $\mathcal{O}(n\sqrt{n})$ since the inducing points of the approximation are chosen to scale with $\sqrt{n}$, a choice that is theoretically justified to provide optimal learning rates as shown in [Marteau-Ferey et al. 2019] and [Rudi et al. 2015]

For larger datasets, $n\sim 10^6$, we found that, choosing $m=\sqrt{n}$ and setting a batch size of 10, computation of all Shapley values for all features only took around 1 minute on V100 cards, which are considered good cards but not top of the line.

Reference:
- [Giacomo et al. 2020]:  Kernel methods through the roof: handling billions of points efficiently
- [Marteau-Ferey et al. 2019]: Globally convergent newton methods for ill-conditioned generalised self-concordant losses.
- [Rudi et al. 2015]: Less is more: Nyström computational regularisation.

---

### Author Response · Authors · 2022-08-07
**Messages to reviewers**

Dear reviewers,

We have addressed all of your constructive comments and questions. We would really appreciate if the reviewers could go over our responses and update their evaluations accordingly. We'd be happy to address any remaining questions - but we will only be able to do this until Tuesday 4 pm ET, so we would appreciate comments before then.

Thank you very much.

Best,
Authors

---

### Meta-Review · Area_Chair_dorQ · 2022-08-30

**Recommendation:** Accept
**Confidence:** Certain

**Metareview:**

The authors propose a novel method for calculating Shapley value for kernel-based models. The paper includes both a theoretical analysis and extensive experimental evaluation. A majority of reviewers are in support of accepting the paper and the rebuttal/discussion period helped to clear out (most of) the reviewers' concerns.

**Award:**

No

---

### Decision · Program_Chairs · 2022-09-14

Accept